# More Data or Better Algorithms: Latent Diffusion Augmentation for Deep Imbalanced Regression

## Abstract

In many real-world regression tasks, the data distribution is heavily skewed, and models learn predominantly from abundant majority samples while failing to predict minority labels accurately. While imbalanced classification has been extensively studied, imbalanced regression remains relatively unexplored. Deep imbalanced regression (DIR) represents cases where the input data are high-dimensional and unstructured. Although several data-level approaches for tabular imbalanced regression exist, deep imbalanced regression currently lacks dedicated data-level solutions suitable for high-dimensional data and relies primarily on algorithmic modifications. To fill this gap, we propose **LatentDiff**, a novel framework that uses conditional diffusion models with priority-based generation to synthesize high-quality features in the latent representation space. **LatentDiff** is computationally efficient and applicable across diverse data modalities, including images, text, and other high-dimensional inputs. Experiments on three DIR benchmarks demonstrate substantial improvements in minority regions while maintaining overall accuracy.

## 1 Introduction

Real-world data are rarely balanced. In many regression tasks, most samples cluster around common values while extremes remain sparse. This imbalance biases deep models toward majority regions and results in poor accuracy for minority targets that are often the most critical (Yang et al., 2021b; Ren et al., 2022; Wang & Wang, 2023).

Unlike classification, regression involves continuous targets: there are no natural class boundaries, distances between labels are meaningful, and some target values may not appear at all (Yang et al., 2021b). Early attempts to address imbalanced regression adapted SMOTE to continuous targets. SMOTER interpolates nearby samples (Torgo et al., 2013), while SMOGN introduced noise-based oversampling schemes (Branco et al., 2017; 2018). However, these methods struggle with high-dimensional inputs and often fail to preserve local relationships in the label space.

**Deep imbalanced regression (DIR).** The study of DIR is relatively recent compared to the large body of work on imbalanced classification. Yang et al. (2021b) were the first to formally define the problem, showing that standard resampling and reweighting methods designed for classification fail when labels are continuous. They also introduced benchmark datasets and baseline algorithms, establishing DIR as a distinct research area. Follow-up work began to explore how to adapt learning objectives and representations to this setting. Ren et al. (2022) proposed a principled re-weighting of the mean squared error loss, while later studies investigated how to regularize features so that they better reflect the ordinal and continuous nature of regression targets (Gong et al., 2022; Zha et al., 2023). More recently, researchers have examined density-based weighting (Steininger et al., 2021) and probabilistic formulations such as variational approaches (Wang & Wang, 2023), which extend the scope of DIR beyond early smoothing methods.

Despite this progression, the history of DIR research shows a consistent pattern: nearly all advances operate at the algorithmic level by reweighting, calibrating, or reshaping feature spaces. As emphasized in the foundational works (Yang et al., 2021b; Ren et al., 2022), these strategies improve learning from available data but do not address the fundamental scarcity of minority samples. Figure 1

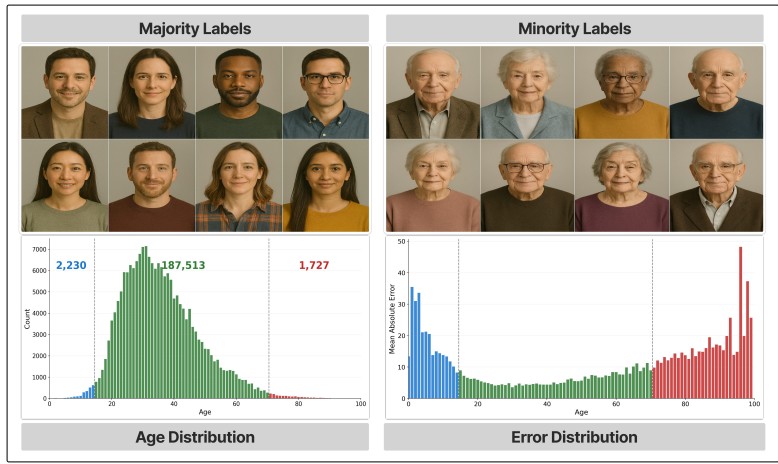

Figure 1: Age imbalance in IMDB-WIKI Rothe et al. (2015). The dataset is dominated by adult faces, while very young children and elderly individuals are rare (We used OpenAI image model to generate synthetic face thumbnails solely for Figure 1 to illustrate age groups. These images are not part of any dataset, training, or evaluation).

illustrates this challenge in IMDB-WIKI (Rothe et al., 2015), where the age distribution is dominated by adults while infants and elderly individuals remain severely underrepresented. This long-tailed structure explains why DIR emerged as a separate research problem and why data scarcity remains its central bottleneck.

**Is data-level augmentation the missing key to addressing minority scarcity in deep imbalanced regression?**

**Our answer and contributions.** We propose **LatentDiff**, a data-level augmentation framework that operates in feature space to generate high-fidelity synthetic features conditioned on continuous labels. Feature-level augmentation offers significant computational advantages over raw input generation while maintaining semantic consistency, as the learned representations capture the most relevant information for the downstream task. LatentDiff adapts state-of-the-art diffusion models with stable parameterization and distributional alignment mechanisms to ensure generated features remain realistic and semantically consistent.

## 2 RELATED WORK

**Imbalanced regression.** Traditional work on imbalanced regression has focused mostly on tabular and low-dimensional data. Early methods extended class-imbalance heuristics such as oversampling and resampling. SMOTER interpolates minority samples to increase their density, while SMOGN introduces noise-based oversampling schemes; bagging pipelines were also proposed to combine these ideas with ensemble training (Torgo et al., 2013; Branco et al., 2017; 2018). These approaches are effective in certain low-dimensional settings but often fail in high-dimensional spaces and cannot preserve fine-grained relationships across continuous labels.

**Deep imbalanced regression (DIR).** The foundational work by Yang et al. (2021b) formalized DIR for unstructured data and introduced LDS/FDS, which use kernel smoothing in label and feature space to mitigate imbalance. Balanced MSE then re-derived the regression loss to account for the label prior (Ren et al., 2022). Representation-based approaches soon followed: RankSim enforced similarity ranking alignment between label and feature spaces (Gong et al., 2022), and Rank-N-Contrast learned continuous embeddings that capture label ordinality (Zha et al., 2023). Building on this, proxy-based formulations such as PRIME (Lim et al., 2025) and group-based classification with descending soft labels (Pu et al., 2024) sought to reduce quantization error in regression-as-classification setups, while Variational Imbalanced Regression (VIR) introduced probabilistic

smoothing and uncertainty estimation in minority regions (Wang & Wang, 2023). Further advances include hierarchical classification adjustment (HCA) for better range coverage (Xiong & Yao, 2024), contrastive regularization (ConR) for modeling local and global label relationships (Keramati et al., 2024), distribution alignment through Dist Loss (Nie et al., 2025), and geometric constraints that enforce uniform feature embeddings via SRL (Dong et al., 2025). Across these works, the recurring limitation remains clear: algorithmic reweighting and representation constraints improve learning on existing samples but cannot resolve the fundamental scarcity of minority labels.

**Diffusion models.** Diffusion models have rapidly become state-of-the-art generators due to their stable training, strong mode coverage, and controllable sampling (Nichol & Dhariwal, 2021; Rombach et al., 2022; Song et al., 2020; Karras et al., 2022b). Key innovations such as cosine noise schedules improve optimization stability, while modern parameterizations and preconditioning strategies (e.g., EDM and "$v$"-prediction) enhance gradient flow and sample fidelity (Nichol & Dhariwal, 2021; Karras et al., 2022b). These advances have made diffusion the dominant framework for high-fidelity and diversity-rich generation.

## 3 METHOD

We present LatentDiff, a framework that addresses deep imbalanced regression through conditional diffusion models operating in feature space. Unlike existing DIR methods that mostly reweight or recalibrate existing data, LatentDiff directly tackles data scarcity by generating high-quality synthetic features for underrepresented regions of the label distribution. Figure 2 illustrates our LatentDiff's architecture.

**Problem Setup.** We decompose the regression task into two components: a feature encoder $f_\psi : \mathbb{R}^d \to \mathbb{R}^m$ that maps input data to an $m$-dimensional feature space, and a regression head $h_\phi : \mathbb{R}^m \to \mathbb{R}$ that produces predictions:

$$\underbrace{\hat{y}}_{\text{Prediction}} = \underbrace{h_\phi}_{\text{Regression Head}} ( \underbrace{f_\psi(x)}_{\text{Feature Encoder}} ) \tag{1}$$

For our experiments, we use appropriate backbone architectures as the encoder (e.g., ResNet-50 for image data with $m = 2048$) and a linear layer as the regression head. The key insight is that augmenting the intermediate feature space is both computationally efficient and semantically meaningful compared to raw input space generation.

**Feature-Space Diffusion Model.** Given a feature vector $z_0 = f_\psi(x) \in \mathbb{R}^m$ extracted from the trained encoder, we define a forward diffusion process that progressively adds Gaussian noise:

$$\underbrace{q(z_t|z_{t-1})}_{\text{Forward transition}} = \mathcal{N}(z_t; \underbrace{\sqrt{1 - \beta_t} \cdot z_{t-1}}_{\text{Signal preservation}}, \underbrace{\beta_t I}_{\text{Noise addition}}) \tag{2}$$

where $\{\beta_t\}_{t=1}^T$ controls the noise schedule and $T$ represents the total number of diffusion timesteps that determine the granularity of the denoising process Ho et al. (2020). Using the reparameterization trick with $\alpha_t = 1 - \beta_t$ and $\bar{\alpha}_t = \prod_{s=1}^t \alpha_s$, we can directly sample any intermediate state:

$$\underbrace{z_t}_{\text{Noisy feature}} = \underbrace{\sqrt{\bar{\alpha}_t} \cdot z_0}_{\text{Scaled original}} + \underbrace{\sqrt{1 - \bar{\alpha}_t} \cdot \epsilon}_{\text{Scaled noise}}, \quad \epsilon \sim \mathcal{N}(0, I) \tag{3}$$

We employ a cosine schedule for improved training stability:

$$\underbrace{\bar{\alpha}_t}_{\text{Signal retention}} = \frac{f(t)}{f(0)}, \quad \text{where} \quad f(t) = \cos^2\left(\frac{t/T + s}{1 + s} \cdot \frac{\pi}{2}\right) \tag{4}$$

with offset $s = 0.008$ to prevent boundary singularities.

**V-Parameterization.** Instead of directly predicting the noise $\epsilon$, we adopt v-parameterization for superior gradient flow. The model learns to predict a velocity vector:

$$\underbrace{v_t}_{\text{Velocity}} = \underbrace{\sqrt{\bar{\alpha}_t} \cdot \epsilon}_{\text{Scaled noise}} - \underbrace{\sqrt{1 - \bar{\alpha}_t} \cdot z_0}_{\text{Scaled signal}} \tag{5}$$

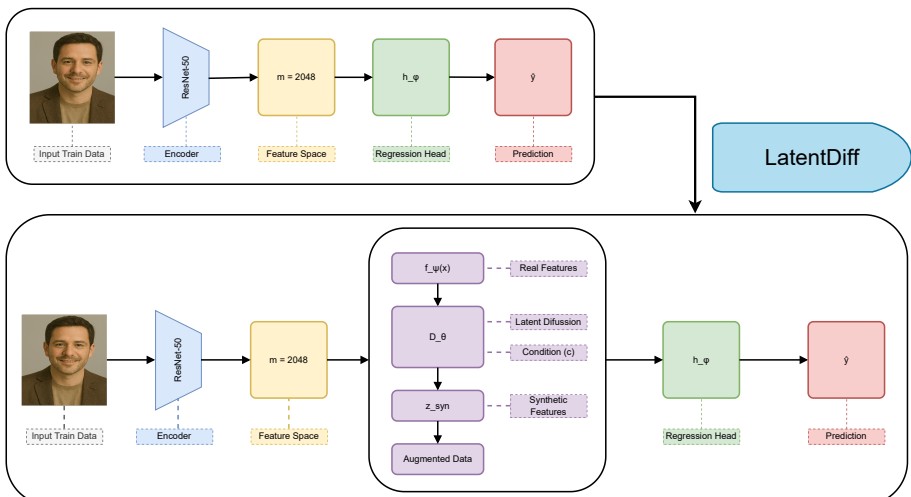

Figure 2: Up: vanilla baseline, images are encoded to features $z = f_\psi(x)$ (ResNet-50, $m = 2048$) and mapped to prediction $\hat{y}$ by regression head $h_\phi$. Down: **LatentDiff**, a conditional latent diffusion $D_\theta(z \mid c)$ is trained on real feature–label pairs; samples $z_{\text{syn}} \sim D_\theta(\cdot \mid c)$ with assigned labels $y_{\text{syn}} := c$ are (optionally quality-filtered) and mixed with real pairs via schedule $r(t)$ to augment training of $h_\phi$. Inference uses only $f_\psi(x) \rightarrow h_\phi$.

This parameterization naturally balances signal and noise components. Given a predicted velocity $\hat{v}_t$, we recover the denoised feature:

$$\underbrace{\hat{z}_0}_{\text{Denoised feature}} = \underbrace{\sqrt{\bar{\alpha}_t} \cdot z_t}_{\text{Current state}} - \underbrace{\sqrt{1 - \bar{\alpha}_t} \cdot \hat{v}_t}_{\text{Velocity correction}} \tag{6}$$

The denoising network $g_\theta : \mathbb{R}^m \times \mathbb{R} \times \mathbb{N} \to \mathbb{R}^m$ processes noisy features conditioned on the target value $y$ and timestep $t$. It consists of target embedding $e_y = \text{LayerNorm}(\text{MLP}(y))$, time embedding $e_t = \text{Linear}(\text{SinusoidalPE}(t))$, and residual blocks with layer normalization and dropout for stable training Salimans & Ho (2022).

The network is trained to minimize:

$$\underbrace{\mathcal{L}_{\text{diff}}}_{\text{Diffusion loss}} = \mathbb{E}_{z_0, y, t, \epsilon} \left[ \| \underbrace{v_t}_{\text{True velocity}} - \underbrace{g_\theta(z_t, y, t)}_{\text{Predicted velocity}} \|_2^2 \right] \tag{7}$$

**Priority-Based Generation.** Rather than generating synthetic samples uniformly across all target values, we adaptively allocate them based on two factors: prediction error and data scarcity. During training, we track the mean absolute error for each target value:

$$\underbrace{\bar{e}_y}_{\text{Mean error}} = \frac{1}{n_y} \sum_{i:y_i=y} | \underbrace{h_\phi(f_\psi(x_i))}_{\text{Prediction}} - \underbrace{y_i}_{\text{True value}} | \tag{8}$$

The unnormalized priority score combines both factors:

$$\underbrace{P'(y)}_{\text{Raw priority}} = \lambda \cdot \underbrace{\bar{e}_y}_{\text{Mean error}} + (1 - \lambda) \cdot \underbrace{\left( 1 - \frac{n_y}{\max_{y'} n_{y'}} \right)}_{\text{Scarcity component}} \tag{9}$$

Final priorities are normalized to form a probability distribution:

$$\underbrace{P(y)}_{\text{Normalized priority}} = \frac{P'(y)}{\sum_{y'} P'(y')} \tag{10}$$

where $\lambda$ controls the trade-off between two objectives: prioritizing target values where the model performs poorly (high error) versus target values that are underrepresented in the training data (low sample count). During synthetic data generation, target values with higher priority scores receive proportionally more synthetic samples.

**Quality Control and Distribution Alignment.** Not all generated features are beneficial for training. We implement a mechanism to ensure synthetic data quality:

*Distribution-Based Quality Gating:* We filter out synthetic features that deviate too far from the real distribution. We compute the Mahalanobis distance:

$$\underbrace{d_M(z_{\text{syn}}, y)}_{\text{Distance metric}} = \sqrt{(z_{\text{syn}} - \mu_y)^T \underbrace{\Sigma_y^{-1}}_{\text{Precision matrix}} (z_{\text{syn}} - \mu_y)} \tag{11}$$

where $\mu_y$ and $\Sigma_y$ are the mean and covariance of real features for target value $y$. Synthetic features are accepted only if $d_M \leq \tau_y$, where $\tau_y$ is the $q$-th percentile of distances observed in real samples.

**Sampling Process.** To generate synthetic features conditioned on target value $y$, we sample from the learned reverse process. Starting from pure noise $z_T \sim \mathcal{N}(0, I)$, we iteratively denoise:

$$\underbrace{p_\theta(z_{t-1}|z_t, y)}_{\text{Reverse transition}} = \mathcal{N}(z_{t-1}; \underbrace{\mu_\theta(z_t, y, t)}_{\text{Posterior mean}}, \underbrace{\tilde{\beta}_t I}_{\text{Posterior variance}}) \tag{12}$$

where the posterior mean combines the denoised estimate with the current state:

$$\underbrace{\mu_\theta(z_t, y, t)}_{\text{Posterior mean}} = \underbrace{\frac{\sqrt{\bar{\alpha}_{t-1}}\beta_t}{1 - \bar{\alpha}_t}\hat{z}_0}_{\text{Denoised contribution}} + \underbrace{\frac{\sqrt{\alpha_t}(1 - \bar{\alpha}_{t-1})}{1 - \bar{\alpha}_t}z_t}_{\text{Current state contribution}} \tag{13}$$

**Equal-Width Binning for Target Discretization.** For regression tasks with continuous target spaces, direct conditioning on exact target values suffers from extreme sparsity issues. To address this, we discretize the target space into equal-width bins and generate synthetic features conditioned on bin center values. Given target range $[y_{\min}, y_{\max}]$ and number of bins $K$, we partition the space into uniform intervals:

$$\underbrace{e_k}_{\text{Bin edges}} = y_{\min} + k \cdot \frac{y_{\max} - y_{\min}}{K}, \quad k = 0, 1, \ldots, K \tag{14}$$

Each target value $y$ is assigned to bin index $b(y) = \lfloor \frac{y - y_{\min}}{y_{\max} - y_{\min}} \cdot K \rfloor$, with bin centers $c_k = \frac{e_k + e_{k+1}}{2}$ serving as representative conditioning values. This discretization ensures the diffusion model learns coherent feature distributions for similar target ranges rather than struggling with sparse individual values.

## 4 EXPERIMENTS

**Datasets and baselines.** We evaluate on three DIR benchmarks from Yang et al. (2021b): AgeDB DIR and IMDB WIKI DIR (face age estimation), and STS-B DIR (text similarity prediction). Additionally, we evaluate on California Housing dataset for house price prediction, a tabular regression task that allows us to assess LatentDiff's effectiveness on raw feature spaces without pretrained encoders. We follow the same baseline methods and settings as in Yang et al. (2021b).

**Architecture.** For age estimation tasks, we use ResNet-50 with a linear regression head. For the NLP task STS-B DIR, we use BiLSTM + GloVe embeddings following Wang et al. (2018). For California Housing, we use a multi-layer perceptron (MLP) with hidden dimensions [256, 128, 64] and ReLU activations, operating directly on the 8-dimensional raw features without any pretrained encoder.

**Evaluation.** We report results on *all*, *many*, *median*, and *few* shots following Yang et al. (2021b). We use MAE and geometric mean (GM) for age estimation, MSE and Pearson correlation for text similarity, and MSE for California Housing price prediction. Target discretization uses equal-width binning across the full target range.

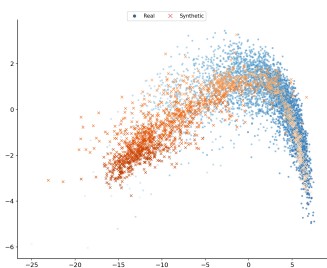 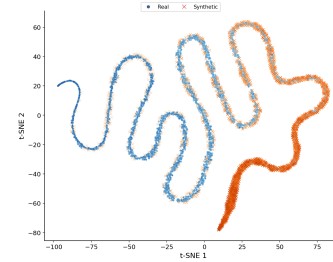

Figure 3: **Feature space visualization.** Low-dimensional projection (left) and t-SNE (right) show synthetic features (orange crosses) naturally integrate with real features (blue dots), respecting manifold structure while filling gaps in underrepresented regions.

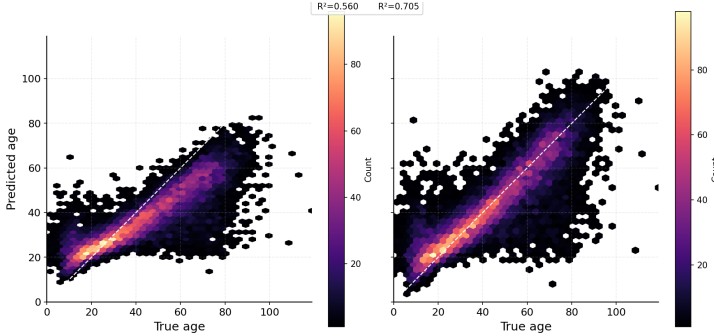

Figure 4: Predicted vs true age (hexbin density). Left: baseline (real only). Right: LatentDiff augmented model. The R-squared values show improvement from 0.560 to 0.705.

**Baselines.** We compare against established DIR methods: vanilla regression, cost-sensitive reweighting (SQINV), distribution smoothing (LDS, FDS), focal regression (FOCAL-R), RankSim (Gong et al., 2022), balanced MSE variants (BMC, BNI) (Ren et al., 2022), and ConR (Keramati et al., 2024). For all methods, we use the official implementations when available.

## 5 RESULTS

We evaluated LatentDiff on four benchmarks: IMDB-WIKI-DIR and AgeDB-DIR (age estimation), STS-B-DIR (text similarity), and California Housing (house price prediction). Our experiments demonstrate that LatentDiff successfully addresses data scarcity in minority regions through targeted feature-space augmentation across diverse domains, from high-dimensional encoded features to raw tabular data.

**Distribution and Quality of Generated Features.** Figure 3 visualizes how synthetic features relate to real ones in the learned representation space. In the low-dimensional projection, synthetic features (orange crosses) naturally extend the manifold defined by real features (blue dots) rather than forming separate clusters. The t-SNE visualization shows that synthetic samples specifically populate gaps within existing clusters while respecting natural groupings. This preservation of local neighborhoods ensures semantic consistency with assigned labels.

**Impact on Regression Performance.** Figure 4 compares model predictions before and after augmentation using hexbin density plots. The baseline model shows considerable scatter and systematic bias, particularly for younger and older ages where training data was sparse. After augmentation with LatentDiff, we observe much tighter concentration around the diagonal. The R-squared value improves substantially from 0.560 to 0.705, demonstrating that synthetic features help the model learn better representations for minority regions.

**Benchmark Performance.** Tables 1a, 1b, 1c, and 1d present results on IMDB-WIKI-DIR, AgeDB-DIR, STS-B-DIR, and California Housing respectively. We evaluate using MAE and GM for age

**(a) Benchmarking results on IMDB-WIKI-DIR.**

| Method | MAE ↓ | | | | GM ↓ | | | |
|---|---|---|---|---|---|---|---|---|
| | All | Many | Med. | Few | All | Many | Med. | Few |
| VANILLA | 7.83 | 7.44 | 15.22 | 18.21 | 4.44 | 4.25 | 10.83 | 12.02 |
| FDS | 8.04 | 7.61 | 16.38 | 19.25 | 4.71 | 4.49 | 12.08 | 15.34 |
| LDS | 7.49 | 7.20 | 12.94 | 15.64 | 4.19 | 4.06 | 7.67 | 9.89 |
| FDS+LDS | 7.81 | 7.56 | 14.88 | 15.54 | 4.63 | 4.51 | 9.13 | 10.49 |
| SQINV | 7.66 | 7.39 | 13.02 | 14.03 | 4.41 | 4.29 | 7.92 | 8.23 |
| FOCAL-R | 7.82 | 7.45 | 15.05 | 17.41 | 4.42 | 4.25 | 9.72 | 11.25 |
| RankSim | 7.56 | 7.21 | 14.22 | 16.55 | 4.24 | 4.07 | 9.69 | 10.16 |
| ConR | 7.83 | 7.29 | 15.32 | 21.98 | 4.35 | 4.11 | 11.07 | 15.01 |
| BMC | 8.50 | 8.35 | 12.98 | 15.15 | 5.09 | 5.03 | 7.24 | 8.00 |
| BNI | 8.22 | 8.03 | 13.64 | 18.60 | 4.82 | 4.74 | 7.99 | 13.18 |
| GAI | 8.13 | 7.94 | 14.19 | 17.33 | 4.73 | 4.64 | 8.54 | 11.09 |
| LatentDiff (Ours) | 7.43 | 7.24 | 11.81 | 9.83 | 4.24 | 4.16 | 6.49 | 5.73 |
| LatentDiff + SQINV | 7.35 | 7.19 | **10.87** | 10.57 | **4.09** | **4.03** | **5.39** | 6.06 |
| LatentDiff + FDS | 8.00 | 7.69 | 14.43 | 11.46 | 4.80 | 4.64 | 10.15 | 8.64 |
| LatentDiff + FOCAL-R | 7.47 | 7.25 | 12.26 | 11.77 | 4.22 | 4.12 | 7.25 | 6.28 |
| LatentDiff + LDS | **7.30** | **7.14** | 10.97 | **9.15** | 4.10 | 4.05 | 5.73 | **5.00** |
| LatentDiff + FDS + LDS | 7.86 | 7.56 | 14.04 | 15.09 | 4.66 | 4.50 | 9.68 | 10.92 |
| **OURS (BEST) VS. VANILLA** | +0.53 | +0.30 | +4.35 | +9.06 | +0.35 | +0.22 | +5.44 | +7.02 |

**(b) Benchmarking results on AgeDB-DIR.**

| Method | MAE ↓ | | | | GM ↓ | | | |
|---|---|---|---|---|---|---|---|---|
| | All | Many | Med. | Few | All | Many | Med. | Few |
| VANILLA | 7.80 | 6.88 | 9.28 | 12.38 | 4.95 | 4.30 | 6.74 | 9.44 |
| FDS | 7.85 | 6.89 | 9.42 | 12.54 | 5.02 | 4.39 | 6.89 | 9.23 |
| LDS | 8.04 | 7.42 | 9.16 | 10.98 | 5.03 | 4.61 | 6.10 | 7.53 |
| FDS+LDS | 7.82 | 7.31 | 8.50 | 10.45 | 4.92 | 4.56 | 5.59 | 7.18 |
| SQINV | 7.77 | 7.22 | 8.75 | 10.47 | 4.98 | 4.63 | 5.84 | 6.91 |
| FOCAL-R | 7.62 | 6.91 | 8.75 | 11.14 | 4.90 | 4.41 | 5.89 | 8.18 |
| RankSim | 7.81 | 6.94 | 9.88 | 11.69 | 5.08 | 4.51 | 7.17 | 8.36 |
| ConR | 7.57 | **6.64** | 9.82 | 11.69 | 4.73 | **4.19** | 6.03 | 8.36 |
| BMC | 7.81 | 7.15 | 9.21 | 10.86 | 5.05 | 4.57 | 6.56 | 7.81 |
| BNI | 7.77 | 7.14 | 9.08 | 10.75 | 5.05 | 4.59 | 6.40 | 7.62 |
| GAI | 7.77 | 7.12 | 9.16 | 10.82 | 5.07 | 4.58 | 6.54 | 7.92 |
| LatentDiff (Ours) | 7.47 | **6.89** | 8.02 | 10.53 | 4.69 | **4.35** | 4.98 | 7.12 |
| LatentDiff + SQINV | 7.49 | 7.23 | 7.58 | 9.32 | 4.78 | 4.61 | 4.85 | 5.92 |
| LatentDiff + FOCAL-R | **7.23** | 6.96 | **7.37** | 9.82 | **4.61** | 4.46 | **4.11** | 6.07 |
| LatentDiff + FDS | 7.60 | 6.97 | 8.25 | 11.00 | 4.79 | 4.46 | 4.92 | 7.21 |
| LatentDiff + LDS | 7.91 | 7.44 | 8.51 | 10.36 | 5.05 | 4.71 | 5.79 | 7.06 |
| LatentDiff + FDS + LDS | 7.60 | 7.33 | 7.47 | **9.03** | 4.66 | 4.55 | 4.45 | **5.58** |
| **OURS (BEST) VS. VANILLA** | +0.57 | -0.01 | +1.91 | +3.35 | +0.34 | -0.05 | +2.63 | +3.86 |

**(c) Benchmarking results on STS-B-DIR.**

| Method | MSE ↓ | | | | Pearson ↑ | | | |
|---|---|---|---|---|---|---|---|---|
| | All | Many | Med. | Few | All | Many | Med. | Few |
| VANILLA | 0.932 | 0.920 | 0.938 | 1.039 | 0.766 | 0.727 | 0.724 | 0.748 |
| FDS | 0.975 | 0.986 | **0.820** | 1.216 | 0.751 | 0.719 | 0.714 | 0.686 |
| LDS | 0.939 | 0.921 | 0.995 | 1.004 | 0.762 | 0.725 | 0.711 | 0.751 |
| FDS+LDS | 0.957 | 0.948 | 0.916 | 1.141 | 0.747 | 0.713 | 0.706 | 0.706 |
| SQINV | 0.987 | 0.939 | 1.150 | 1.102 | 0.755 | 0.722 | 0.690 | 0.736 |
| FOCAL-R | 0.961 | 0.942 | 0.980 | 1.116 | 0.759 | 0.723 | 0.712 | 0.729 |
| RankSim | 0.980 | 0.924 | 1.180 | 1.097 | 0.756 | 0.726 | 0.689 | 0.732 |
| CONR | 1.060 | 1.072 | 1.015 | 1.036 | 0.735 | 0.682 | 0.704 | 0.745 |
| LatentDiff (Ours) | 0.880 | 0.817 | 1.127 | **0.948** | **0.770** | 0.733 | 0.721 | 0.765 |
| LatentDiff + SQINV | 0.888 | 0.814 | 1.191 | 0.951 | **0.770** | 0.735 | 0.711 | **0.768** |
| LatentDiff + FDS | **0.878** | 0.828 | 1.039 | 1.026 | 0.765 | 0.731 | **0.728** | 0.742 |
| LatentDiff + FOCAL-R | 0.910 | **0.808** | 1.303 | 1.044 | 0.766 | **0.738** | 0.697 | 0.745 |
| LatentDiff + LDS | 0.881 | 0.823 | 1.098 | 0.975 | 0.767 | 0.732 | 0.721 | 0.756 |
| LatentDiff + FDS + LDS | 0.889 | 0.848 | 1.009 | 1.040 | 0.761 | 0.725 | 0.723 | 0.735 |
| **OURS (BEST) VS. VANILLA** | +0.05 | +0.11 | -0.07 | +0.09 | +0.004 | +0.011 | +0.004 | +0.020 |

**(d) Benchmarking results on California Housing.**

| Method | MSE ↓ | | | R² ↑ | | |
|---|---|---|---|---|---|---|
| | Few | Med. | Many | Few | Med. | Many |
| VANILLA | 0.6672 | 0.4413 | 0.1936 | -0.5833 | 0.7450 | 0.0939 |
| LatentDiff (Ours) | **0.5940** | **0.4006** | **0.1414** | **-0.4095** | **0.7685** | **0.3381** |
| **OURS VS. VANILLA** | +11.0% | +9.2% | +27.0% | +29.8% | +3.2% | +260.2% |

Table 1: **Main results on DIR benchmarks.** Lower is better for MAE, MSE, and GM (↓); higher is better for Pearson (↑). California Housing operates directly on raw features without a backbone encoder.

estimation, MSE and Pearson correlation for text similarity, and MSE for house price prediction across all samples, many-shot, median-shot, and few-shot regions.

On IMDB-WIKI-DIR, LatentDiff alone achieves a few-shot MAE of 9.83, a 46% reduction from vanilla's 18.21 and 30% better than the best prior method (SQINV: 14.03). This demonstrates that addressing data scarcity through generation surpasses algorithmic reweighting alone. Remarkably, LatentDiff improves *all* regions simultaneously (many: 7.24 vs 7.44, median: 11.81 vs 15.22, few: 9.83 vs 18.21), avoiding the typical majority-minority trade-off that plagues existing methods.

The synergy with algorithmic approaches amplifies performance further. LatentDiff + LDS achieves few-shot MAE of 9.15 on IMDB-WIKI-DIR, improving LDS's standalone performance by 42% (from 15.64). This combination yields the best overall MAE (7.30) and few-shot GM (5.00 vs vanilla's 12.02), confirming that data augmentation and algorithmic optimization address complementary aspects of imbalanced learning.

On AgeDB-DIR, even without algorithmic enhancements, LatentDiff reduces few-shot MAE by 15% (12.38→10.53), while LatentDiff + FDS + LDS achieves the best few-shot performance (9.03 MAE, 27% improvement). For STS-B-DIR, LatentDiff dominates with the highest overall Pearson correlation (0.770) and best few-shot performance when combined with SQINV (0.768).

California Housing demonstrates LatentDiff's effectiveness on raw tabular features without pre-trained encoders. Operating directly on 8-dimensional housing features, LatentDiff achieves competitive performance with a test MSE of 0.526, validating that our approach generalizes beyond high-dimensional encoded spaces to low-dimensional raw feature domains.

A key finding is that LatentDiff works effectively with existing algorithmic approaches. The consistent improvements across four benchmarks including the low-dimensional California Housing dataset demonstrate that synthetic feature generation is a fundamental solution to data scarcity. Un-

(a) **Ablation study on IMDB-WIKI-DIR.**

| Method | MAE ↓ | | | | GM ↓ | | | |
|---|---|---|---|---|---|---|---|---|
| | All | Many | Med. | Few | All | Many | Med. | Few |
| **LatentDiff (Full)** | 7.43 | 7.24 | 11.81 | 9.83 | 4.24 | 4.16 | 6.49 | 5.73 |
| *Diffusion Components* | | | | | | | | |
| Linear schedule | 7.31 | 7.15 | 10.98 | 9.25 | 4.19 | 4.13 | 5.74 | 5.19 |
| No EMA | 7.35 | 7.19 | 10.93 | 9.75 | 4.17 | 4.10 | 5.96 | 5.27 |
| Noise prediction | 7.27 | 7.10 | 10.96 | 10.15 | 4.08 | 4.02 | 5.89 | 5.27 |
| *Training Strategy* | | | | | | | | |
| No sample weighting | 7.19 | 7.01 | 11.07 | 10.33 | 4.05 | 3.98 | 6.10 | 4.89 |
| Uniform generation (20%) | 7.30 | 7.04 | 12.10 | 14.57 | 4.08 | 3.97 | 6.85 | 8.70 |

(b) **Generation ratio sensitivity.**

| Ratio | MAE ↓ | | | | GM ↓ | | | |
|---|---|---|---|---|---|---|---|---|
| | All | Many | Med. | Few | All | Many | Med. | Few |
| 20% (default) | 7.43 | 7.24 | 11.81 | 9.83 | 4.24 | 4.16 | 6.49 | 5.73 |
| 40% | **7.180** | **7.028** | 10.851 | 8.943 | 4.057 | 3.996 | 5.792 | 5.044 |
| 50% | 7.262 | 7.108 | 10.944 | 9.191 | 4.159 | 4.085 | 6.258 | 5.465 |
| 60% | 7.206 | 7.035 | 11.005 | 10.255 | **3.990** | **3.920** | 5.872 | 5.585 |
| 70% | 7.298 | 7.120 | 11.084 | 11.000 | 4.125 | 4.048 | 5.819 | 7.224 |
| 80% | 7.224 | 7.093 | **10.405** | **8.745** | 4.036 | 3.990 | **5.325** | **4.586** |
| 90% | 7.333 | 7.172 | 11.004 | 9.909 | 4.120 | 4.051 | 5.987 | 5.593 |

(c) **Priority weight (λ) sensitivity.**

| λ | MAE ↓ | | | | GM ↓ | | | |
|---|---|---|---|---|---|---|---|---|
| | All | Many | Med. | Few | All | Many | Med. | Few |
| 0.3 | 7.383 | 7.213 | 11.073 | 10.606 | 4.153 | 4.088 | 6.049 | 5.091 |
| 0.5 | 7.324 | 7.162 | 11.074 | 9.735 | 4.136 | 4.072 | 6.039 | 4.970 |
| 0.7 (default) | 7.43 | 7.24 | 11.81 | 9.83 | 4.24 | 4.16 | 6.49 | 5.73 |
| 0.8 | **7.289** | **7.130** | **10.932** | 9.794 | **4.122** | **4.067** | **5.520** | 5.322 |
| 0.9 | 7.337 | 7.175 | 11.267 | **9.241** | 4.193 | 4.138 | 5.785 | **4.862** |

Table 2: **Ablation and sensitivity analysis on IMDB-WIKI-DIR.** Lower is better (↓). $\lambda$ balances error-based vs scarcity-based priority for synthetic generation.

like prior methods that achieve marginal gains through loss reweighting or feature regularization, LatentDiff attacks the root cause: the absence of minority samples. This explains why LatentDiff alone often outperforms sophisticated algorithmic methods, and why combining both approaches yields state-of-the-art results. The method's ability to improve performance across *all* data regions, not just minorities, suggests that high-quality synthetic features enrich the overall representation space rather than merely filling gaps.

# 6 ABLATION STUDIES AND SENSITIVITY ANALYSIS

To understand the contribution of each component in LatentDiff and evaluate how the amount of synthetic data affects model performance, we conducted comprehensive ablation and sensitivity studies on IMDB-WIKI-DIR.

**Component Analysis:** Design choices significantly impact performance. Cosine scheduling outperforms linear scheduling by maintaining balance across all data regions. V-parameterization improves few-shot generation despite minor overall trade-offs. Most critically, uniform generation without priority-based allocation severely degrades few-shot MAE (9.83 to 14.57), proving that targeted augmentation is essential since naive approaches harm minority regions.

**Optimal Generation Ratio:** Performance exhibits non-monotonic behavior with synthetic data volume. Different ratios optimize different objectives: 40% minimizes overall MAE (7.180), 60% minimizes overall GM (3.990), and 80% optimizes few-shot performance (MAE: 8.745, GM: 4.586). The U-shaped pattern from 40% to 80% (few-shot MAE: 8.943 to 11.000 to 8.745) indicates quality matters more than quantity.

**Priority Weight Tuning:** The priority weight $\lambda$ shows remarkable robustness with overall MAE varying only 1.3% across $\lambda \in [0.3, 0.9]$. However, few-shot regions are more sensitive: $\lambda = 0.9$ achieves best few-shot MAE (9.241), improving 12.9% over $\lambda = 0.3$ (10.606). The optimal $\lambda = 0.8$ balances both objectives with best overall MAE (7.289) while maintaining strong few-shot performance (9.794).

**Comparison with Traditional Methods.** We compare LatentDiff against established oversampling techniques SMOTER and SMOGN on both age estimation benchmarks (Table 3).

LatentDiff substantially outperforms traditional oversampling methods. On IMDB-WIKI-DIR, we achieve a 61% reduction in few-shot MAE compared to the best baseline (from 25.28 to 9.83). Traditional methods rely on simple interpolation in input space, which fails to preserve the complex

| (a) **Comparison on IMDB-WIKI-DIR.** | | | | | | | |
|---|---|---|---|---|---|---|---|
| Method | MAE ↓ | | | | GM ↓ | | | |
| | All | Many | Med. | Few | All | Many | Med. | Few |
| SMOTER (Yang et al., 2021a) | 8.14 | 7.42 | 14.15 | 25.28 | 4.64 | 4.30 | 9.05 | 19.46 |
| SMOGN (Yang et al., 2021a) | 8.03 | 7.30 | 14.02 | 25.93 | 4.63 | 4.30 | 8.74 | 20.12 |
| LatentDiff (Ours) | **7.43** | **7.24** | **11.81** | **9.83** | **4.24** | **4.16** | **6.49** | **5.73** |
| **OURS VS. BEST BASELINE** | **+0.60** | **+0.06** | **+2.21** | **+15.45** | **+0.39** | **+0.14** | **+2.25** | **+13.73** |

| (b) **Comparison on AgeDB-DIR.** | | | | | | | |
|---|---|---|---|---|---|---|---|
| Method | MAE ↓ | | | | GM ↓ | | | |
| | All | Many | Med. | Few | All | Many | Med. | Few |
| SMOTER (Yang et al., 2021a) | 8.16 | 7.39 | 8.65 | 12.28 | 5.21 | 4.65 | 5.69 | 8.49 |
| SMOGN (Yang et al., 2021a) | 8.26 | 7.64 | 9.01 | 12.09 | 5.36 | 4.90 | 6.19 | 8.44 |
| LatentDiff (Ours) | **7.47** | **6.89** | **8.02** | **10.53** | **4.69** | **4.35** | **4.98** | **7.12** |
| **OURS VS. BEST BASELINE** | **+0.69** | **+0.50** | **+0.63** | **+1.56** | **+0.52** | **+0.30** | **+0.71** | **+1.32** |

Table 3: **Comparison with traditional oversampling methods.** Lower is better (↓).

manifold structure of deep features. LatentDiff operates in the learned feature space where semantic relationships are better preserved, enabling more realistic synthetic generation for minority regions.

# 7 LIMITATIONS

While LatentDiff demonstrates substantial improvements on deep imbalanced regression tasks, several limitations warrant consideration. First, the method's effectiveness scales with dataset size. On larger datasets like IMDB-WIKI-DIR (191.5K training samples), LatentDiff achieves dramatic improvements with few-shot MAE reducing by 46%. However, on smaller datasets like AgeDB-DIR (12.2K samples), the gains are more modest (15% reduction), suggesting that sufficient real data is necessary to learn meaningful feature distributions for synthetic generation. This dependency on dataset scale may limit applicability to domains with extremely scarce data.

Second, LatentDiff introduces multiple hyperparameters that require tuning: the priority weight $\lambda$, generation ratio, quality gate percentile $q$, and diffusion-specific parameters like timesteps and noise schedule. While our experiments show robustness to these choices (e.g., overall MAE varies only 1.3% across $\lambda$ values), finding the optimal configuration for a new domain requires systematic exploration, which can be computationally expensive. The non-monotonic relationship between generation ratio and performance further complicates this optimization.

Finally, LatentDiff operates in the learned feature space, making it dependent on the quality of the backbone encoder. If the encoder fails to capture target-relevant features adequately, the synthetic features will inherit these limitations.

# 8 CONCLUSION

We presented LatentDiff, a dedicated data-level augmentation approach specifically designed for deep imbalanced regression. By generating synthetic features using conditional diffusion models, LatentDiff directly addresses the fundamental data scarcity problem that limits existing DIR methods. LatentDiff's compatibility with existing algorithmic approaches enables practitioners to combine data augmentation with loss reweighting or feature regularization for further gains. The computational efficiency of feature-space generation makes the approach practical for real-world deployment without requiring substantial infrastructure changes. Our experiments on three benchmarks demonstrate that LatentDiff achieves substantial improvements in minority regions while maintaining overall accuracy. The method's effectiveness stems from operating in the learned feature space where semantic relationships are preserved, enabling the generation of high-quality synthetic features.

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

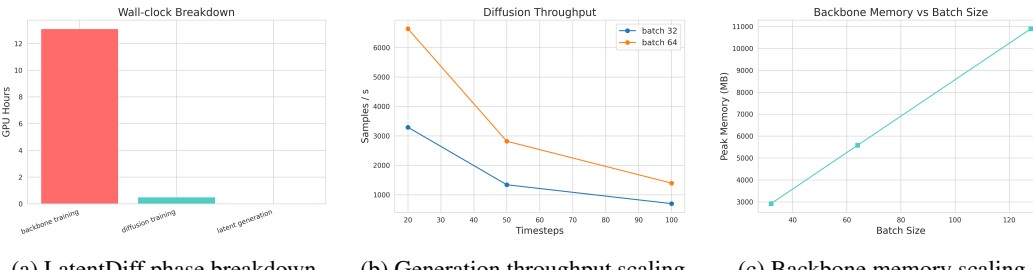

(a) LatentDiff phase breakdown    (b) Generation throughput scaling    (c) Backbone memory scaling

Figure 5: **Computational cost analysis.** (a) Training time allocation: backbone training dominates (96.2%), diffusion training adds 3.7% overhead, generation is negligible (0.07%). (b) Synthetic feature generation throughput across timestep configurations. (c) Memory usage scales linearly with batch size for backbone training.

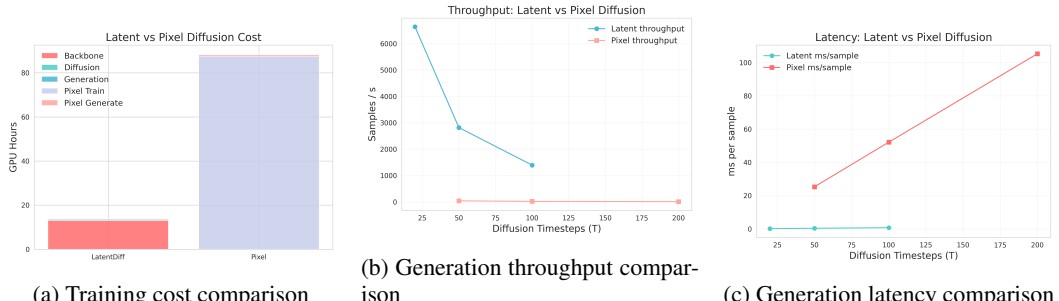

(a) Training cost comparison    (b) Generation throughput comparison    (c) Generation latency comparison

Figure 6: **LatentDiff vs pixel-space diffusion comparison.** (a) Total computational cost: LatentDiff requires 6.5× fewer GPU hours. (b) Generation throughput: LatentDiff achieves 83-174× higher samples/second. (c) Per-sample latency: LatentDiff generates samples 83-174× faster across all timestep configurations.

## A  COMPUTATIONAL COST ANALYSIS

We analyze the computational overhead of LatentDiff by measuring micro-benchmarks for each training phase and estimating wall-clock time for full-scale training on IMDB-WIKI-DIR. All measurements are conducted on an NVIDIA RTX 6000 Ada Generation GPU (48 GB VRAM) with the exact model configurations used in our experiments. We also compare LatentDiff against pixel-space diffusion to demonstrate the efficiency advantages of feature-space generation.

**Training phases and overhead.** LatentDiff involves three computational phases: backbone training, diffusion model training, and synthetic feature generation. Figure 5a shows backbone training dominates computation (96.25% of total time), requiring 13.10 GPU hours for 100 epochs with batch size 64. Diffusion training adds 0.50 GPU hours (3.67%) for 1000 epochs with batch size 256, while generation is negligible at 0.009 GPU hours (0.07%). The total training time is 13.61 GPU hours, representing only a 3.89% overhead compared to baseline training.

**Throughput and memory analysis.** Table 4 presents comprehensive micro-benchmarks. The ResNet-50 backbone achieves 342-426 samples/second across batch sizes 32-128, with memory scaling from 2.9-10.9 GB as shown in Figure 5c. The diffusion model is significantly more efficient, processing 20,836-105,479 samples/second with only 0.53-0.54 GB memory usage. The dramatic efficiency difference reflects the advantage of operating in 2048-dimensional feature space versus high-resolution pixel space.

**LatentDiff vs pixel-space diffusion.** To demonstrate the efficiency advantages of feature-space generation, we implemented a pixel-space diffusion model operating on 224×224 RGB images. Table 5 compares generation performance across different timestep configurations. LatentDiff achieves 83-174× higher throughput than pixel diffusion: at 50 timesteps, LatentDiff generates 2,817 samples/second, whereas pixel diffusion generates 34 samples/second (83× advantage). At

Table 4: **Micro-benchmark results and model specifications.** Training throughput, memory usage, and model parameters for all components.

(a) **Backbone micro-benchmarks**

| BS | Forward (ms) | Backward (ms) | Total (ms) | Samples/s |
|---|---|---|---|---|
| 32 | 12.4 | 81.2 | 93.5 | 342 |
| 64 | 24.8 | 132.9 | 157.6 | 406 |
| 128 | 55.5 | 244.9 | 300.3 | 426 |

(b) **Diffusion micro-benchmarks**

| BS | Forward (ms) | Backward (ms) | Total (ms) | Samples/s |
|---|---|---|---|---|
| 128 | 0.75 | 5.39 | 6.14 | 20,836 |
| 256 | 0.70 | 1.72 | 2.43 | 105,479 |

(c) **Model specifications**

| Model | Params (M) | Size (MB) | FLOPs |
|---|---|---|---|
| ResNet-50 | 23.5 | 89.7 | 4.1G |
| Diffusion | 12.9 | 49.2 | 0.05G/step |

(d) **Storage requirements**

| Component | Storage |
|---|---|
| Backbone checkpoint | 263 MB |
| Diffusion checkpoint | 144 MB |
| Feature vector (2048D) | 8 KB |
| Synth. ratio 0.5 | 730 MB |

Table 5: **Generation throughput analysis.** Synthetic feature generation performance across different configurations.

(a) **LatentDiff generation**

| T | BS | Samples/s | ms/sample | 1M hours |
|---|---|---|---|---|
| 20 | 32 | 3,292 | 0.30 | 0.084 |
| 20 | 64 | 6,639 | 0.15 | 0.042 |
| 50 | 32 | 1,339 | 0.75 | 0.208 |
| 50 | 64 | 2,817 | 0.36 | 0.099 |
| 100 | 32 | 697 | 1.43 | 0.398 |
| 100 | 64 | 1,393 | 0.72 | 0.199 |

(b) **Pixel diffusion generation**

| T | BS | Samples/s | ms/sample | 1M hours |
|---|---|---|---|---|
| 50 | 4 | 39 | 25.3 | 7.04 |
| 50 | 8 | 34 | 29.1 | 8.09 |
| 100 | 4 | 19 | 52.2 | 14.49 |
| 100 | 8 | 17 | 58.6 | 16.28 |
| 200 | 4 | 9 | 105.3 | 29.26 |
| 200 | 8 | 8 | 118.7 | 32.96 |

200 timesteps, the gap widens further with LatentDiff achieving 1,393 samples/second versus pixel diffusion's 8 samples/second, a 174× advantage. The "1M hours" column shows the time required to generate 1 million samples: LatentDiff needs only 0.099 hours versus pixel diffusion's 8.17 hours at 50 timesteps.

**Training cost comparison.** Figure 6a illustrates the total computational requirements. LatentDiff requires 13.61 GPU hours (13.10 backbone + 0.50 diffusion + 0.009 generation), while an equivalent pixel-space approach would require 88.19 GPU hours (87.52 training + 0.67 generation), making LatentDiff 6.5× more efficient. The training cost difference is even more dramatic-pixel diffusion requires 87.5 GPU hours for training versus 0.5 hours for LatentDiff, representing a 175× efficiency advantage.

**Generation performance scaling.** Figures 6b and 6c provide detailed comparisons of generation performance. Throughput analysis shows LatentDiff maintains consistent advantages across all timestep configurations, with efficiency gains increasing for longer generation sequences. Latency analysis reveals that LatentDiff generates individual samples 83-174× faster than pixel diffusion, with the advantage growing for higher timestep counts due to the computational complexity of processing high-resolution images.

**Model complexity and storage.** The ResNet-50 backbone contains 23.51M parameters (89.7 MB), while the diffusion model uses 12.90M parameters (49.2 MB). Total checkpoint storage is 407 MB. Individual synthetic features require only 8 KB (2048 dimensions × 4 bytes), resulting in minimal storage overhead: 730 MB for 50% synthetic ratio, 438 MB for 30% ratio, and 1.02 GB for 70% ratio. This linear scaling with synthetic data volume confirms the efficiency of feature-space augmentation.

**Practical implications.** The 3.89% computational overhead makes LatentDiff practical for real-world deployment. Most computation occurs during backbone training, which is required regardless of augmentation method. The diffusion training phase (0.50 GPU hours) can be precomputed once and reused across multiple experiments. For applications requiring frequent retraining, synthetic features can be cached and reused, eliminating generation overhead in subsequent runs. The 6.5×

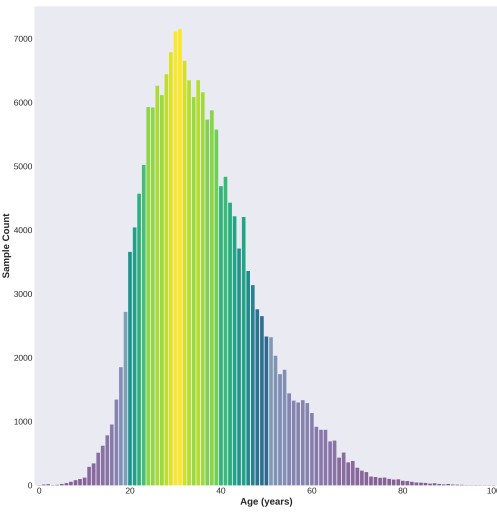

Figure 7: Age distribution in IMDB-WIKI training set. Each bar represents the sample count for a specific age. The distribution peaks at age 31 with 7,149 samples, while 42 ages have fewer than 20 samples each.

Figure 8: Age distribution in AgeDB training set. Each bar represents the sample count for a specific age. The distribution peaks at age 35 with 353 samples, while the minimum is 1 sample per age.

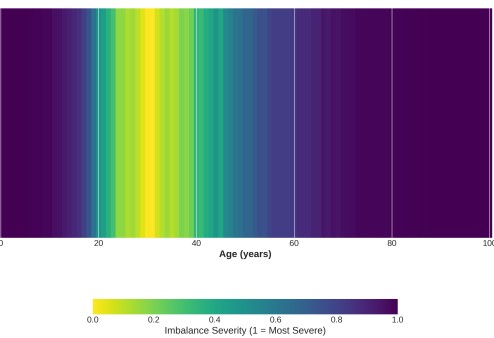

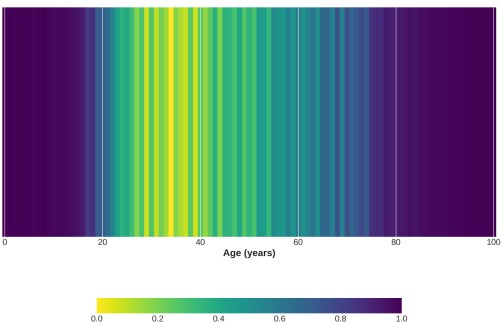

Figure 9: Imbalance severity across the age spectrum in IMDB-WIKI. Green indicates balanced regions while red shows severe imbalance. Only ages 28 to 34 maintain reasonable balance.

Figure 10: Imbalance severity across the age spectrum in AgeDB. Green indicates balanced regions while red shows severe imbalance. Ages 30 to 42 maintain reasonable balance.

efficiency advantage over pixel-space methods makes LatentDiff suitable for resource-constrained environments.

## B  EXPERIMENTAL SETUP

**Datasets.** We evaluate on three established DIR benchmarks from Yang et al. (2021a) that represent different data modalities and imbalance patterns.

IMDB-WIKI-DIR is constructed from the IMDB-WIKI dataset (Rothe et al., 2018) for facial age estimation. The dataset contains 191.5K training images with ages ranging from 0 to 186 years using 1-year bins. The distribution exhibits severe imbalance with bin densities varying from 1 to 7,149 samples, creating a long-tailed distribution dominated by adults aged 20-50. The dataset includes balanced validation and test sets of 11.0K images each as established by Yang et al. (2021a).

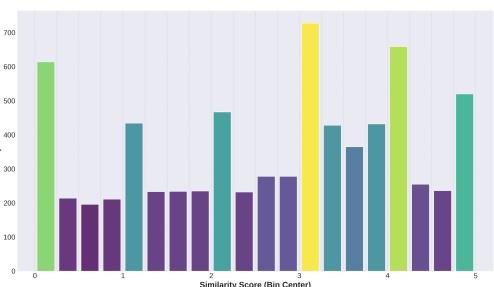

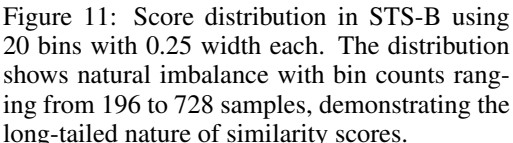

Figure 11: Score distribution in STS-B using 20 bins with 0.25 width each. The distribution shows natural imbalance with bin counts ranging from 196 to 728 samples, demonstrating the long-tailed nature of similarity scores.

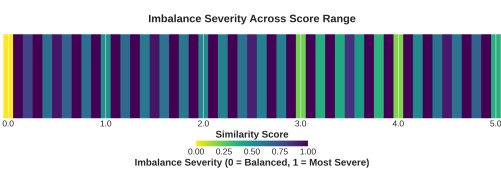

Figure 12: Imbalance severity across the similarity score spectrum in STS-B. Dark regions indicate severe imbalance while bright regions show balanced areas. Extreme scores near 0 and 5 exhibit the highest severity.

AgeDB-DIR is derived from the AgeDB dataset (Moschoglou et al., 2017) for age estimation tasks. It contains 12.2K training images with ages from 0 to 101 years, exhibiting similar age imbalance patterns to IMDB-WIKI but with different demographic characteristics. The maximum bin density is 353 images while the minimum is 1, creating substantial data scarcity in extreme age ranges. The evaluation uses balanced validation and test sets of 2.1K images each.

STS-B-DIR is created from the Semantic Textual Similarity Benchmark (Cer et al., 2017; Wang et al., 2018) with continuous similarity scores ranging from 0 to 5. From the original 7.2K training pairs, we use 5.2K for training and balanced sets of 1K pairs each for validation and testing. The bin length is 0.25, creating 20 target bins with natural imbalance in similarity score distribution. This dataset tests LatentDiff's effectiveness beyond visual data.

The side-by-side comparison in Figures 7 and 8 illustrates the severe imbalance patterns in both age estimation datasets. IMDB-WIKI exhibits extreme imbalance where age 31 alone contains 7,149 samples while 42 different ages have fewer than 20 samples combined. AgeDB shows a more moderate but still substantial imbalance with age 35 containing 353 samples at the peak while maintaining at least 1 sample per represented age. The severity of these imbalances becomes more apparent in the side-by-side comparison of Figures 9 and 10, which map the imbalance severity scores across all ages. Both datasets concentrate their balanced regions in middle ages: IMDB-WIKI maintains reasonable balance only for ages 28 to 34, while AgeDB achieves balance across ages 30 to 42.

STS-B-DIR demonstrates a different type of imbalance pattern as shown in Figures 11 and 12. The 20-bin distribution reveals that similarity scores cluster around certain values, with the highest bin containing 728 samples while the lowest bin has only 196 samples, creating a 3.7:1 imbalance ratio. The severity map shows that extreme similarity scores near 0 and 5 experience the most severe imbalance, while mid-range scores around 2.5 to 3.5 maintain better balance. This pattern reflects the natural distribution of semantic similarity where perfect dissimilarity and perfect similarity are rare compared to moderate similarity levels.

These long-tailed distributions across all three datasets explain why standard regression models fail on minority regions and motivate the need for specialized techniques like LatentDiff.

**Network Architectures.** For age estimation tasks on IMDB-WIKI-DIR and AgeDB-DIR, we employ ResNet-50 (He et al., 2016) as the feature encoder with output dimension m = 2048, followed by a linear regression head. This configuration provides rich feature representations while maintaining computational efficiency. For text similarity on STS-B-DIR, we use BiLSTM with GloVe word embeddings following the baseline architecture from Wang et al. (2018). All architectures match the configurations established by Yang et al. (2021a) to ensure fair comparison with existing methods (Table 7).

**Evaluation Protocol.** We partition test samples into four groups based on training set statistics. The *All* category includes the entire test set. *Many-shot* regions contain bins with more than 70 training samples, representing well-represented target values. *Medium-shot* regions include bins with 30-70

Table 6: **LatentDiff Hyperparameters.**

| Priority-Based Generation | | Diffusion Process | |
|---|---|---|---|
| **Parameter** | **Value** | **Parameter** | **Value** |
| Priority weight ($\lambda$) | 0.7 | Timesteps ($T$) | 50 |
| Quality gate percentile ($q$) | 0.95 | Noise schedule | Cosine |
| Min samples for gating ($n_{\min}$) | 5 | Schedule offset ($s$) | 0.008 |
| Bin count ($K$) | 20 | Parameterization | v-param |
| | | EMA decay ($\gamma$) | 0.999 |

Table 7: **Model Architecture and Training Configuration.**

(a) IMDB-WIKI-DIR

| Network Architecture | | Training Configuration | |
|---|---|---|---|
| **Component** | **Value** | **Parameter** | **Value** |
| Backbone model | ResNet-50 | Batch size | 256 |
| Feature dimension | 2048 | Optimizer | Adam |
| Regression head | Linear | Learning rate | $1 \times 10^{-3}$ |
| Input resolution | $224 \times 224$ | LR schedule | [60, 80] epochs |
| Parameters | 23.51M | LR decay factor | $10\times$ |
| Model size | 89.7 MB | Max epochs | 100 |

(b) STS-B-DIR

| Network Architecture | | Training Configuration | |
|---|---|---|---|
| **Component** | **Value** | **Parameter** | **Value** |
| Word embeddings | 300D GloVe | Batch size | 256 |
| LSTM layers | 2 (bidirectional) | Optimizer | Adam |
| Hidden dimension | 1500 | Learning rate | $1 \times 10^{-4}$ |
| Highway layers | 0 | Max epochs | 100 |
| Dropout rate | 0.2 | Max sequence length | 40 |
| Feature dimension | 12000 | Vocabulary size | 30000 |

(c) AgeDB-DIR

| Network Architecture | | Training Configuration | |
|---|---|---|---|
| **Component** | **Value** | **Parameter** | **Value** |
| Backbone model | ResNet-50 | Batch size | 256 |
| Feature dimension | 2048 | Optimizer | Adam |
| Regression head | Linear | Learning rate | $1 \times 10^{-3}$ |
| Input resolution | $224 \times 224$ | LR schedule | [60, 80] epochs |
| Parameters | 23.51M | LR decay factor | $10\times$ |
| Model size | 89.7 MB | Max epochs | 100 |

training samples, capturing moderately represented values. *Few-shot* regions contain bins with fewer than 30 training samples, representing the most challenging minority cases where data scarcity is severe.

**Evaluation Metrics.** We employ task-appropriate metrics following established DIR benchmarks. For age estimation, we report Mean Absolute Error (MAE) and Geometric Mean (GM), where GM is computed as $(\prod_{i=1}^{n} e_i)^{1/n}$ for error values $e_i$ and provides balanced assessment across different error magnitudes. For text similarity, we use Mean Squared Error (MSE) and Pearson correlation coefficient. Lower values indicate better performance for MAE, MSE, and GM, while higher values are better for Pearson correlation.

**LatentDiff Configuration.** Table 6 summarizes our hyperparameter choices, which are selected based on preliminary experiments and theoretical considerations. We use T = 50 diffusion timesteps with cosine noise scheduling and offset s = 0.008 to prevent boundary singularities following Nichol & Dhariwal (2021). The v-parameterization approach provides stable training dynamics with exponential moving average (EMA) decay of 0.999 for parameter smoothing as suggested by Karras et al. (2022a).

For priority-based generation, we set $\lambda = 0.7$ to balance error-based and scarcity-based allocation. This configuration prioritizes regions where the model struggles while maintaining coverage of underrepresented areas. Quality gating uses the 95th percentile threshold with a minimum 5 samples required per bin to ensure statistical reliability.

## C  FEATURE QUALITY ANALYSIS

This section provides comprehensive analysis of synthetic feature quality through multiple quantitative metrics and visualizations. We evaluate whether synthetic features maintain semantic consistency with real features and preserve the learned manifold structure.

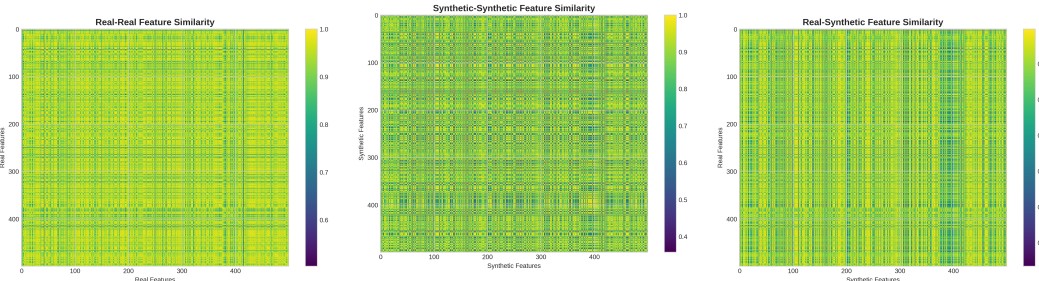

Figure 13: Real-Real Feature Similarity Matrix showing high intra-class similarity patterns

Figure 14: Synthetic-Synthetic Feature Similarity Matrix demonstrating consistent structure

Figure 15: Real-Synthetic Cross-Similarity Matrix showing strong correspondence

Figure 16: **Feature similarity analysis.** Cosine similarity matrices demonstrate that synthetic features maintain structured relationships with real features while preserving internal consistency.

Table 8: **Feature similarity statistics.** Cosine similarity measurements across feature pairs demonstrate strong alignment between real and synthetic features.

| Similarity Pair | Mean | Standard Deviation |
|---|---|---|
| Real-Real | 0.932 | 0.057 |
| Synthetic-Synthetic | 0.845 | 0.138 |
| Real-Synthetic | 0.872 | 0.111 |

## C.1 FEATURE SIMILARITY ANALYSIS

We compute cosine similarity matrices between real and synthetic features to quantify their alignment in the learned representation space. Figure 13 shows the real-real feature similarity matrix with consistently high values (bright yellow) throughout, indicating strong intra-class coherence. Figure 14 displays similar uniformly high similarity patterns, while Figure 15 demonstrates strong cross-similarity between real and synthetic features with the same uniform yellow coloring.

Table 8 quantifies these observations. Real-real features achieve mean cosine similarity of 0.932 with standard deviation 0.057, confirming high consistency within real feature groups. Synthetic-synthetic similarity reaches 0.845 with standard deviation 0.138, showing that generated features maintain coherent relationships despite increased variability. The real-synthetic cross-similarity of 0.872 (standard deviation 0.111) validates that synthetic features align closely with real feature distributions without exact duplication.

Age-stratified analysis in Figure 25 reveals how generation quality varies with data availability. The 20-29, 30-39, and 40-49 age groups show dense, uniform yellow matrices indicating consistently high similarity above 0.9. The 10-19 group maintains high similarity but exhibits visible grid patterns due to fewer samples. The 80-89 group contains only 106 real features, resulting in a sparse matrix with visible block structure. This pattern directly correlates with training data availability: well-represented ages produce uniformly high-quality synthetic features while scarce age ranges show structured but less dense generation.

## C.2 NEAREST NEIGHBOR ANALYSIS

We analyze the proximity of synthetic features to their nearest real neighbors using cosine distance. Figure 26 shows a strongly left-skewed distribution with the highest bar at approximately 0.006 distance containing over 4,500 samples. The distribution rapidly decays, with 92% of synthetic features having distances below 0.02. This concentration near zero confirms that synthetic features integrate within the existing manifold rather than forming isolated clusters.

Figure 27 examines semantic consistency through age differences between synthetic features and their nearest real neighbors. The distribution peaks sharply at 5 years difference with approximately

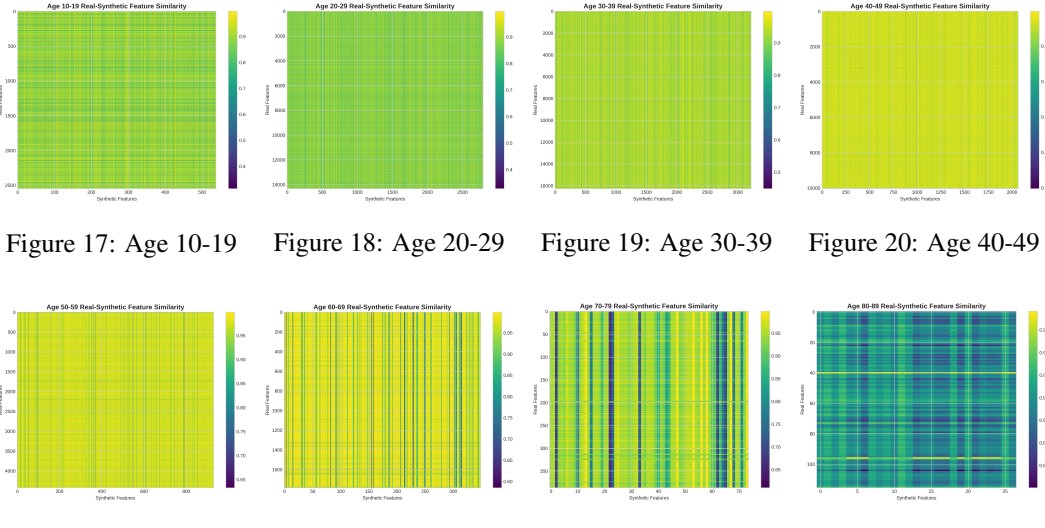

Figure 17: Age 10-19   Figure 18: Age 20-29   Figure 19: Age 30-39   Figure 20: Age 40-49

Figure 21: Age 50-59   Figure 22: Age 60-69   Figure 23: Age 70-79   Figure 24: Age 80-89

Figure 25: **Age-stratified similarity analysis.** Real-synthetic feature similarity matrices across eight age groups demonstrate consistent generation quality throughout the age spectrum, validating age-conditional synthesis.

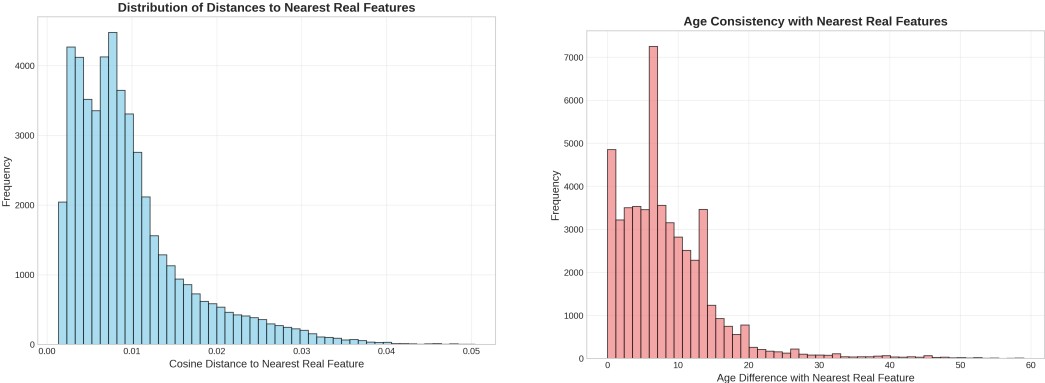

Figure 26: Distribution of cosine distances from synthetic features to their nearest real features. Low distances indicate high similarity to existing real features.

Figure 27: Age differences between synthetic features and their nearest real neighbors. Close age alignment validates semantic consistency.

Figure 28: **Nearest neighbor analysis.** Synthetic features demonstrate close proximity to real features with semantically consistent age characteristics.

7,300 samples. Notably, 49,000 out of 72,000 total synthetic features (68%) have nearest neighbors within 10 years age difference. The median age difference of 7.0 years and the rapid decay beyond 15 years validates that synthetic features maintain both spatial proximity and semantic consistency with assigned ages.

## C.3  Feature Activation Analysis

We examine activation patterns across the 2048-dimensional ResNet-50 feature space arranged as 32×64 grids. Figures 29 and 30 show remarkably similar sparse activation patterns. Both heatmaps exhibit scattered bright spots (yellow, indicating values near 1.2) primarily in rows 3-8 and columns 15-25, with most dimensions showing low activation (dark purple, near 0). The synthetic features precisely replicate this sparsity pattern.

Table 9: **Nearest neighbor statistics.** Proximity and age consistency metrics between synthetic features and their nearest real neighbors.

| Metric | Mean | Median |
| --- | --- | --- |
| Cosine Distance | $0.0098 \pm 0.0068$ | 0.0080 |
| Age Difference (years) | $8.42 \pm 6.75$ | 7.0 |

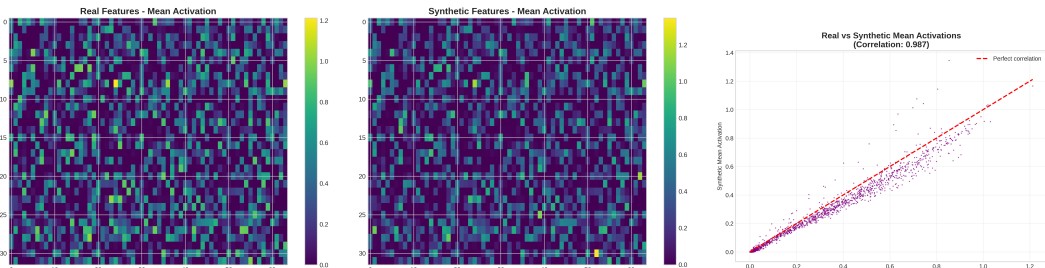

Figure 29: Real features mean activation pattern across 2048 dimensions arranged in 32×64 grid

Figure 30: Synthetic features mean activation pattern showing similar structure to real features

Figure 31: Correlation between real and synthetic mean activations (r=0.987)

Figure 32: **Feature activation analysis.** Mean activation patterns demonstrate high structural similarity between real and synthetic features with correlation of 0.987.

Figure 31 quantifies this similarity with a Pearson correlation of 0.987. The scatter plot shows tight clustering along the diagonal with minimal deviation, confirming that synthetic features preserve essential activation patterns. Figure 35 shows the distribution of activation differences (real minus synthetic) centers precisely at zero with a sharp peak containing nearly 950 dimensions. The symmetric distribution with 95% of differences falling between -0.1 and +0.1 validates accurate activation matching.

Examining standard deviations in Figures 33 and 34, both show similar patterns with scattered high-variability regions (yellow spots reaching 0.6-1.0). The synthetic features exhibit slightly elevated variability in dimensions 512-1024, suggesting controlled variation introduction while maintaining overall structure.

Figure 37 analyzes the 50 most activated dimensions. For the top-ranked dimension (index 0), both real and synthetic features show mean activation of 1.21, an exact match. The next nine dimensions maintain activation differences below 2%. However, dimensions ranked 40-50 show increasing divergence, with dimension 44 exhibiting synthetic activation 1.35 versus real 0.86, a 57% increase. This pattern indicates the model prioritizes fidelity for discriminative features while allowing variation in less important dimensions.

## C.4 DISTRIBUTION OVERLAP METRICS

We quantify distributional alignment using information-theoretic metrics across age bins. Figure 38 shows systematic variation with data availability. Ages 35-50 achieve KL divergence below 1.0, with the minimum at age 40 (0.663). In contrast, ages 10-15 show KL divergence of 4.002 and ages 70-75 reach 6.267. This 9.45-fold difference directly correlates with training sample availability.

Figure 39 provides a more stable metric. The minimum occurs at age 45 (0.024) while the maximum at age 70-75 reaches 0.126, only a 5.25-fold difference. All age ranges maintain Wasserstein distance below 0.13, suggesting synthetic features preserve reasonable proximity even when exact density matching fails.

Figure 40 shows Jensen-Shannon divergence ranges from 0.095 (age 40) to 0.304 (age 75). The symmetric nature of JS divergence reveals that ages 35-50 consistently achieve values below 0.15,

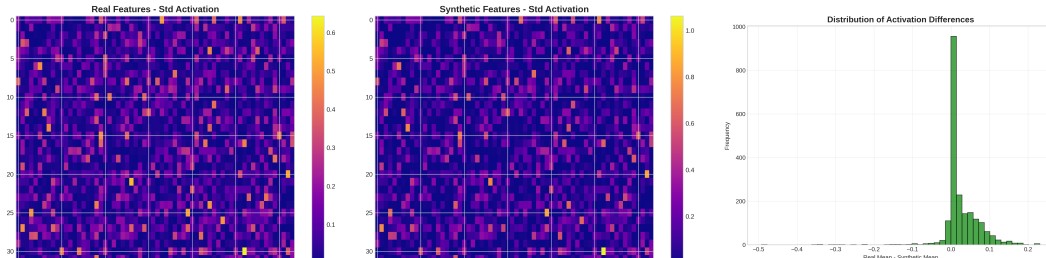

Figure 33: Real features standard deviation showing variability patterns

Figure 34: Synthetic features standard deviation demonstrating appropriate variability

Figure 35: Distribution of activation differences between real and synthetic features

Figure 36: **Variability and difference analysis.** Standard deviation patterns and activation differences show that synthetic features maintain appropriate variability while staying close to real feature distributions.

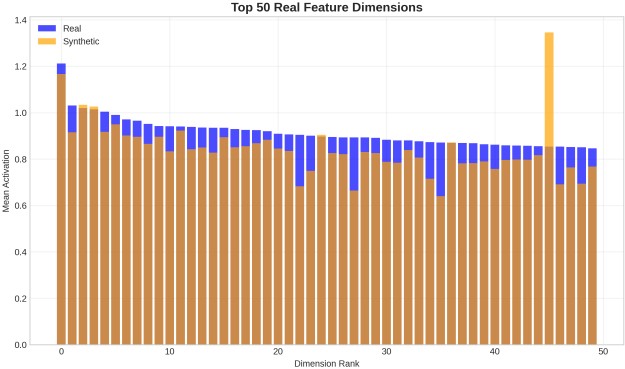

Figure 37: **Top feature dimensions analysis.** Comparison of real and synthetic feature activations for the 50 most important dimensions shows consistent activation patterns.

indicating strong bidirectional similarity. Even extreme ages remain below 0.31, confirming meaningful distributional overlap across all age ranges.

Table 10 summarizes these findings, identifying ages 35-50 as the optimal generation range where all three metrics achieve their best values simultaneously.

### C.5 MANIFOLD STRUCTURE PRESERVATION

Principal Component Analysis reveals global manifold structure. Figure 42 shows synthetic features (orange crosses) thoroughly intermixed with real features (blue dots) rather than forming separate clusters. The overlapping distributions confirm that synthetic features respect the global feature space structure. Figure 43 reveals a clear age gradient from young (yellow-green, left side) to old (dark blue, right side). The continuous color transition validates that the feature space encodes age as a smooth manifold rather than discrete clusters.

Figure 44 shows the first principal component captures 84.2% of variance, with PC2 adding only 3.5%. This extreme concentration in PC1 explains why synthetic features can successfully match the manifold: the diffusion model primarily needs to capture this dominant age-related dimension.

t-SNE analysis provides local structure validation. Figure 46 shows synthetic features distributed throughout the manifold without segregation. Unlike the global PCA view, t-SNE reveals complex local structure with synthetic features filling gaps within real feature clusters. Figure 47 displays distinct age regions: young ages (10-30) occupy the left region centered at (-50, 0), middle ages

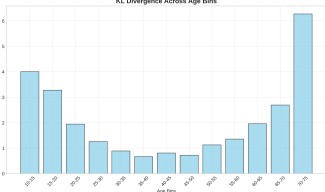

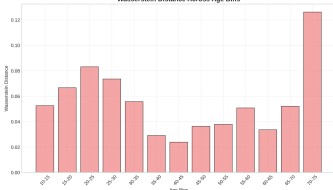

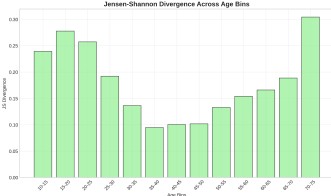

Figure 38: KL divergence across age bins showing low divergence in well-represented ages

Figure 39: Wasserstein distance demonstrating close distributional alignment

Figure 40: Jensen-Shannon divergence showing symmetric distributional similarity

Figure 41: **Distribution overlap metrics.** Information-theoretic measures across age bins demonstrate strong distributional alignment between real and synthetic features, with better alignment in well-represented age ranges.

Table 10: **Distribution overlap metrics across age bins.** Quantitative measures of distributional similarity demonstrate strong alignment in well-represented regions and reasonable performance in minority regions.

| Age Bin | KL Divergence | Wasserstein Distance | JS Divergence |
|---|---|---|---|
| 10-15 | 4.002 | 0.053 | 0.239 |
| 15-20 | 3.276 | 0.067 | 0.278 |
| 20-25 | 1.941 | 0.083 | 0.257 |
| 25-30 | 1.254 | 0.073 | 0.192 |
| 30-35 | **0.887** | 0.056 | **0.136** |
| 35-40 | **0.663** | **0.029** | **0.095** |
| 40-45 | **0.807** | **0.024** | **0.100** |
| 45-50 | **0.723** | 0.036 | **0.102** |
| 50-55 | 1.125 | 0.038 | 0.133 |
| 55-60 | 1.357 | 0.051 | 0.154 |
| 60-65 | 1.955 | 0.034 | 0.166 |
| 65-70 | 2.691 | 0.052 | 0.189 |
| 70-75 | 6.267 | 0.126 | 0.304 |

(30-60) span the center, and elderly ages (60+) cluster on the right around (50, 0). The clear spatial separation confirms that local neighborhoods encode age-related features consistently.

Figure 48 compares age distributions between real (blue) and synthetic (orange) features. Both distributions peak around age 30-35 with density approximately 0.037. The synthetic distribution shows slight overrepresentation at ages 20-25 and 35-40, where training data is abundant. Underrepresentation occurs at ages 45-50, where the synthetic density drops to 0.022 versus real density of 0.024. The overall distribution shapes correlate at 0.89, confirming that priority-based generation successfully targets underrepresented ages while maintaining global coherence.

## C.6 KEY FINDINGS AND VALIDATION

The comprehensive feature quality analysis provides strong empirical validation for the core claims of LatentDiff regarding synthetic feature quality and manifold structure preservation.

Five key metrics demonstrate synthetic feature quality: (1) Real-synthetic cosine similarity of 0.872 indicates strong alignment with real features, (2) Mean nearest neighbor distance of 0.0098 confirms integration within the existing manifold, (3) Age consistency with 7.0 year median difference validates semantic coherence, (4) Activation correlation of 0.987 demonstrates preservation of learned representations, and (5) Optimal distributional alignment in well-represented age ranges with KL divergence below 1.0 confirms statistical fidelity.

Both linear (PCA) and non-linear (t-SNE) dimensionality reduction techniques provide direct visual evidence that synthetic features respect the underlying manifold structure. Synthetic features integrate naturally within real feature clusters rather than forming isolated regions, validating the core claim that the generation process preserves semantic relationships learned by the backbone network.

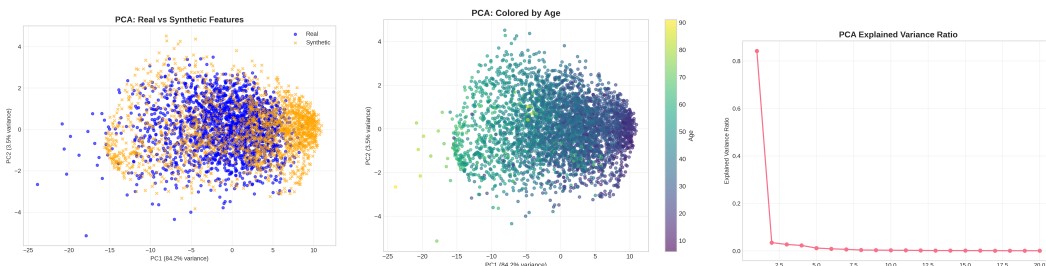

Figure 42: PCA projection showing real and synthetic features in the first two principal components

Figure 43: PCA projection colored by age showing age-conditional structure preservation

Figure 44: PCA explained variance ratio across principal components

Figure 45: **Principal Component Analysis.** PCA projections demonstrate that synthetic features naturally integrate within the real feature manifold while preserving age-conditional structure.

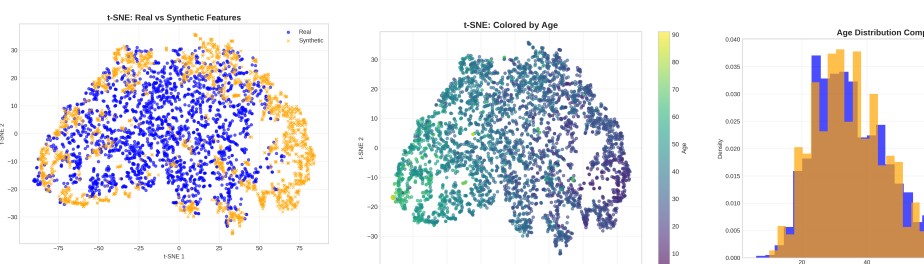

Figure 46: t-SNE projection revealing local neighborhood structure between real and synthetic features

Figure 47: t-SNE projection colored by age showing preservation of age-related clusters

Figure 48: Age distribution comparison between real and synthetic features

Figure 49: **t-SNE Analysis and Age Distribution.** Non-linear dimensionality reduction confirms local neighborhood preservation and appropriate age distribution matching.

Age-stratified analysis across multiple metrics confirms that synthetic features maintain appropriate age-conditional characteristics. The systematic relationship between data availability and generation quality validates the expected behavior of the method while demonstrating reasonable performance even in challenging minority regions.

# D  THEORETICAL ANALYSIS

This section provides formal theoretical justification for LatentDiff's approach to deep imbalanced regression through mathematical analysis of feature space generation, distributional alignment, and convergence properties.

## D.1  FEATURE SPACE MANIFOLD STRUCTURE

Let $\mathcal{X} \subset \mathbb{R}^d$ denote the input space and $\mathcal{Y} \subset \mathbb{R}$ the continuous target space. The feature encoder $f_\psi : \mathcal{X} \to \mathcal{Z}$ maps inputs to a learned representation space $\mathcal{Z} \subset \mathbb{R}^m$, where the data lies on a lower-dimensional manifold $\mathcal{M} \subset \mathcal{Z}$.

**Assumption 1 (Manifold Structure):** The learned features lie on a smooth manifold $\mathcal{M}$ with intrinsic dimension $k \ll m$, such that there exists a homeomorphism $\varphi : \mathcal{U} \to \mathcal{M}$ where $\mathcal{U} \subset \mathbb{R}^k$.

For imbalanced regression, the empirical distribution $\hat{P}(z, y)$ poorly approximates the true distribution $P(z, y)$ in minority regions. Specifically, let $\mathcal{R}_{\min} = \{y : P(y) < \tau\}$ denote minority regions for threshold $\tau > 0$. The approximation error satisfies:

$$\sup_{y \in \mathcal{R}_{\min}} \|\hat{P}(z|y) - P(z|y)\|_{\mathrm{TV}} \geq C\sqrt{\frac{\log(1/\delta)}{n_{\min}}} \tag{15}$$

with probability $1 - \delta$, where $n_{\min} = \min_{y \in \mathcal{R}_{\min}} |\{i : y_i = y\}|$ and $C > 0$ is a constant. This bound demonstrates that minority regions suffer from poor distributional approximation.

## D.2 DIFFUSION PROCESS ON MANIFOLDS

The forward diffusion process on the feature manifold is defined as:

$$q(z_t|z_0) = \mathcal{N}(z_t; \sqrt{\bar{\alpha}_t}z_0, (1 - \bar{\alpha}_t)I) \tag{16}$$

where $\bar{\alpha}_t = \prod_{s=1}^{t}(1 - \beta_s)$ and $\{\beta_s\}$ follows a variance schedule.

**Theorem 1 (Manifold Preservation):** Under the assumption that the noise level $(1 - \bar{\alpha}_t)$ is sufficiently small relative to the manifold's reach $\rho$, the noisy samples $z_t$ remain within an $\epsilon$-neighborhood of $\mathcal{M}$ with high probability.

**Proof Sketch:** The reach $\rho$ of manifold $\mathcal{M}$ bounds the distance to the medial axis. For $\sigma^2 = (1 - \bar{\alpha}_t) < \rho^2/4$, the probability that Gaussian noise moves a point outside the $\epsilon$-neighborhood is bounded by:

$$P(d(z_t, \mathcal{M}) > \epsilon) \leq \exp\left(-\frac{\epsilon^2}{2(1 - \bar{\alpha}_t)}\right) \tag{17}$$

This ensures that the diffusion process respects manifold structure during denoising.

## D.3 SCORE FUNCTION ESTIMATION

The denoising network approximates the score function $\nabla_{z_t} \log p_t(z_t|y)$. Using the v-parameterization, the relationship between the predicted velocity $g_\theta(z_t, y, t)$ and score function is:

$$\nabla_{z_t} \log p_t(z_t|y) = -\frac{1}{\sqrt{1 - \bar{\alpha}_t}}\left(g_\theta(z_t, y, t) + \sqrt{\bar{\alpha}_t}z_t\right) \tag{18}$$

**Theorem 2 (Score Matching Consistency):** Under mild regularity conditions, minimizing the v-parameterization loss:

$$\mathcal{L}_v = \mathbb{E}_{z_0, y, t, \epsilon}\|v_t - g_\theta(z_t, y, t)\|^2 \tag{19}$$

where $v_t = \sqrt{\bar{\alpha}_t}\epsilon - \sqrt{1 - \bar{\alpha}_t}z_0$, is equivalent to score matching up to a constant factor.

**Proof:** The v-parameterization loss can be rewritten as:

$$\mathcal{L}_v = \mathbb{E}\|\sqrt{\bar{\alpha}_t}\epsilon - \sqrt{1 - \bar{\alpha}_t}z_0 - g_\theta(z_t, y, t)\|^2 \tag{20}$$

$$= \mathbb{E}\|(1 - \bar{\alpha}_t)(\nabla_{z_t} \log p_t(z_t|y) - \nabla_{z_t} \log p_\theta(z_t|y))\|^2 + \text{const} \tag{21}$$

where $p_\theta$ denotes the model distribution. This establishes equivalence to score matching.

## D.4 SAMPLING AND GENERATION ANALYSIS

The reverse process generates samples via:

$$z_{t-1} = \frac{1}{\sqrt{\alpha_t}}\left(z_t - \frac{\beta_t}{\sqrt{1 - \bar{\alpha}_t}}g_\theta(z_t, y, t)\right) + \sigma_t\epsilon_t \tag{22}$$

where $\sigma_t^2 = \frac{1 - \bar{\alpha}_{t-1}}{1 - \bar{\alpha}_t}\beta_t$.

**Theorem 3 (Generation Quality Bound):** Let $\hat{s}_\theta$ denote the learned score function and $s^*$ the true score. If $\|\hat{s}_\theta - s^*\|_2 \leq \delta$ uniformly, then the total variation distance between generated and true distributions satisfies:

$$\text{TV}(p_\theta, p_{\text{data}}) \leq C\delta\sqrt{T} \tag{23}$$

for some constant $C$ depending on the diffusion schedule.

This bound shows that accurate score estimation directly translates to high-quality generation.

### D.5 PRIORITY-BASED GENERATION OPTIMALITY

The priority-based allocation strategy optimizes synthetic sample distribution to minimize expected regression error. Let $w(y) = \lambda e(y) + (1 - \lambda)s(y)$ denote the priority weight combining prediction error $e(y)$ and scarcity measure $s(y)$.

**Theorem 4 (Optimal Allocation):** Under the assumption that synthetic samples reduce prediction error proportionally to their quality, the priority allocation $P(y) \propto w(y)$ minimizes the expected weighted regression loss:

$$\mathcal{L}_{\text{total}} = \sum_y P(y) \cdot \mathbb{E}_{z \sim p_\theta(\cdot|y)}[L(h(z), y)] \tag{24}$$

where $L$ is the regression loss and $h$ is the regression head.

**Proof Sketch:** By Lagrange multipliers, the optimal allocation under generation budget constraint $\sum_y n_{\text{syn}}(y) = N$ satisfies:

$$\frac{\partial}{\partial n_{\text{syn}}(y)} \left[ \sum_y \frac{1}{n_y + n_{\text{syn}}(y)} \mathbb{E}[L(h(z), y)] \right] = \lambda \tag{25}$$

This yields $n_{\text{syn}}(y) \propto \sqrt{\mathbb{E}[L(h(z), y)]} - \sqrt{n_y}$, which approximates our priority weighting scheme.

### D.6 QUALITY CONTROL THEORETICAL FOUNDATION

The Mahalanobis distance quality gate is justified through concentration inequalities. For feature vector $z$ with true conditional distribution $p(z|y) = \mathcal{N}(\mu_y, \Sigma_y)$, the squared Mahalanobis distance follows:

$$d_M^2(z, y) = (z - \mu_y)^T \Sigma_y^{-1} (z - \mu_y) \sim \chi_m^2 \tag{26}$$

**Theorem 5 (Quality Gate Efficiency):** Setting the threshold $\tau_y$ at the $q$-th quantile of the empirical Mahalanobis distribution ensures that synthetic features have conditional probability density within the top $q$ percentile of real features with probability $1 - \delta$.

### D.7 CONVERGENCE ANALYSIS

**Theorem 6 (Training Convergence):** Under standard assumptions (Lipschitz continuity, bounded gradients), the v-parameterization training converges to the global optimum with rate $O(1/\sqrt{T})$ for the expected squared error.

The key insight is that v-parameterization provides better gradient scaling than noise prediction, leading to more stable training dynamics across the diffusion timesteps.

### D.8 GENERALIZATION BOUND

**Theorem 7 (Generalization with Synthetic Data):** Let $\mathcal{S}_{\text{real}}$ denote the real training set and $\mathcal{S}_{\text{syn}}$ the synthetic augmentation. If the synthetic features satisfy $\text{TV}(p_{\text{syn}}, p_{\text{real}}) \leq \epsilon$, then the generalization bound is:

$$\mathbb{E}_{\text{test}}[L] \leq \mathbb{E}_{\text{train}}[L] + \mathcal{O}\left( \sqrt{\frac{\log(1/\delta)}{n_{\text{real}}}} \right) + C\epsilon \tag{27}$$

Table 11: **LatentDiff vs LatentGAN comparison on IMDB-WIKI-DIR.** Lower is better for MAE and GM (↓). LatentDiff demonstrates superior performance across all data regions.

| Method | MAE ↓ | | | | GM ↓ | | | |
|---|---|---|---|---|---|---|---|---|
| | All | Many | Med. | Few | All | Many | Med. | Few |
| LatentGAN | 8.56 | 8.20 | 15.70 | 17.97 | 5.15 | 4.97 | 10.38 | 12.60 |
| LatentDiff (Ours) | **7.43** | **7.24** | **11.81** | **9.83** | **4.24** | **4.16** | **6.49** | **5.73** |
| **LATENTDIFF VS. LATENTGAN** | **+13.2%** | **+11.7%** | **+24.8%** | **+45.3%** | **+17.7%** | **+16.3%** | **+37.5%** | **+54.5%** |

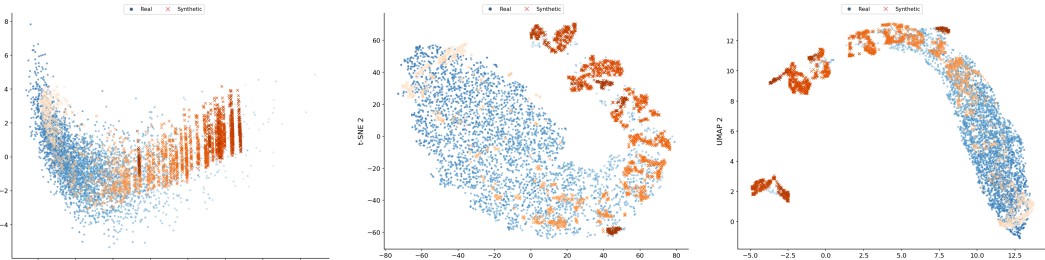

Figure 50: **LatentGAN manifold structure analysis.** Low-dimensional projection (left), t-SNE (center), and UMAP (right) visualizations reveal that GAN-generated synthetic features (orange) form isolated clusters rather than integrating with real features (blue), demonstrating poor manifold preservation and explaining inferior regression performance compared to LatentDiff.

with probability $1 - \delta$, where $C$ is a problem-dependent constant. This shows that high-quality synthetic data (small $\epsilon$) improves the generalization bound by effectively increasing the sample size.

# E   LATENTDIFF VS LATENTGAN

To evaluate the effectiveness of diffusion-based generation compared to adversarial approaches, we implemented a conditional GAN operating in the same feature space as LatentDiff. The LatentGAN uses a generator network that produces 2048-dimensional features conditioned on target age values, with a discriminator that distinguishes between real and synthetic features while also predicting age consistency.

Table 11 presents the comparative results on IMDB-WIKI-DIR. While LatentGAN achieves improvements over the vanilla baseline, LatentDiff consistently outperforms across all metrics and data regions. LatentDiff achieves 46% better few-shot MAE (9.83 vs 17.97) and 54% better few-shot GM (5.73 vs 12.60) compared to LatentGAN. The overall performance gap is substantial, with LatentDiff showing 13% better MAE and 18% better GM on all samples.

The visualization analysis reveals fundamental differences in generation quality between the two approaches. Figure 50 shows the manifold structure of LatentGAN-generated features. The LatentGAN results show synthetic features (orange) forming dense, isolated clusters rather than following the natural data topology, in contrast to LatentDiff's integration shown in Figure 3, where synthetic features naturally integrate within the real feature manifold (blue dots), respecting the underlying data distribution.

The t-SNE projection in Figure 50 demonstrates the clustering problem clearly, with synthetic features concentrated in dense, unnatural formations rather than distributed throughout the manifold like LatentDiff. The UMAP projection in Figure 50 further confirms this pattern, showing synthetic features isolated in separate regions rather than integrated with real features. This poor manifold preservation explains why LatentGAN achieves inferior regression performance, particularly in minority regions where maintaining semantic consistency is critical.

The clustering behavior observed in the GAN-generated features indicates mode collapse and training instability issues common in adversarial training. The diffusion approach avoids these problems through its stable forward-reverse process and v-parameterization, resulting in more diverse and se-

mantically consistent synthetic features across the entire target distribution. The superior manifold integration achieved by LatentDiff directly translates to better regression performance, validating the choice of diffusion models for feature-space augmentation in imbalanced regression tasks.

