# OpenReview forum: "More Data or Better Algorithms: Latent Diffusion Augmentation for Deep Imbalanced Regression"
_ICLR.cc/2026/Conference — ICLR 2026 Conference Withdrawn Submission_

### Official Review · Reviewer_PnTr · 2025-10-29

**Soundness:** 2
**Presentation:** 3
**Contribution:** 2
**Rating:** 4
**Confidence:** 5

**Summary:**

This paper proposes LatentDiff, a method for deep imbalanced regression DIR that uses a conditional diffusion model to generate synthetic features in the latent space for underrepresented target values. The approach includes a priority-based generation mechanism to focus on high-error and low-data regions and a quality-gating step based on Mahalanobis distance. The authors demonstrate LatentDiff's effectiveness on several benchmarks (IMDB-WIKI-DIR, AgeDB-DIR, STS-B-DIR, California Housing), showing significant improvements, particularly in the few-shot regime, and that it can be combined with existing algorithmic methods like LDS.

**Strengths:**

1.  Applying diffusion models to feature-space augmentation for imbalanced regression is a new and timely idea.

2. The paper is well-structured, the method is described in detail, and the experimental investigation is extensive.

3. The writing is clear, and the figures effectively illustrate the method's properties and advantages.

4. The work demonstrates that advanced generative models can be an effective tool for data-level augmentation in DIR.

**Weaknesses:**

The paper's contributions are significantly undermined by the following critical weaknesses:

1. Overstated contribution: The central claim that data-level augmentation is the "missing key" for DIR is inaccurate and ignores established literature. The work fails to cite and benchmark against highly relevant prior works that have already pioneered data-level solutions for DIR, such as:

Zhu et al., "IRDA: Implicit data augmentation for deep imbalanced regression," Information Sciences, 2024. This paper explicitly introduces an implicit data augmentation framework for DIR, operating in the feature space.

Keramati et al., "ConR: Contrastive regularizer for deep imbalanced regression," ICLR, 2024 (mentioned in the paper but the data augmentaition in this paper was not referred to). This method employs problem-specific explicit data augmentations on the input space for DIR. [Although this paper is cited, its data augmentation part is not explicitly mentioned]

Shmuel et al., "Data Augmentation for Deep Learning Regression Tasks by Machine Learning Models," arXiv, Jan, 2025. This work also explores a data-centric approach for regression tasks. By omitting these direct contemporaries, the paper misrepresents the novelty of its core premise.

2. The approach relies on a fixed, pre-trained feature encoder. The diffusion model is trained on features from this static encoder, but the regression head is then trained on a mixture of real and synthetic features. This creates an unaddressed circular dependency and distribution shift, as the synthetic features become invalid as the optimal feature representation evolves during training.

3. The empirical evaluation lacks critical comparisons necessary to validate the claims. Most notably, there is few comparison with previous data augmentation methods for DIR. Furthermore, there is no ablation study replacing the complex diffusion model with a simpler generative model (e.g., a VAE) to prove its necessity over a simpler baseline.

**Questions:**

1. Given the existence of Zhu et al. and Shmuel et al.' studies, which also performs data-level augmentation for DIR, what is the fundamental advancement of your work beyond an incremental model swap?

2. How does the performance and feature quality change if the encoder is fine-tuned alongside the regression head, rather than being kept frozen?

3. Can you provide a direct ablation comparing your diffusion model to a simpler conditional VAE or GAN to conclusively demonstrate its necessity for the performance gains?

---

### Official Review · Reviewer_iZYu · 2025-10-31

**Soundness:** 2
**Presentation:** 1
**Contribution:** 2
**Rating:** 2
**Confidence:** 4

**Summary:**

This paper addresses the problem of Deep Imbalanced Regression, where models perform poorly on underrepresented “minority” target values. The authors argue that most existing solutions are algorithmic (e.g., re-weighting or loss regularization) and fail to solve the key problem of data scarcity.
They propose LatentDiff, a data-level augmentation framework that uses a conditional diffusion model to generate synthetic data in the latent space.  Experiments on three DIR benchmarks and one tabular dataset  show clear improvement in minority regions.

**Strengths:**

- The paper tackles the critical and relevant challenge of Deep Imbalanced Regression, a problem that is far less explored than its classification counterpart.
- The approach is simple and easy to follow.

**Weaknesses:**

1. The feature encoder is trained on the same imbalanced dataset, so its features for minority samples are likely poor and biased. Since the diffusion model is trained on these features, it might simply reproduce or amplify that bias. Could the paper provide a motivation of this design choice ?

2. Generating synthetic points in latent space seems conceptually similar to manifold mixup or interpolation-based augmentation methods. The paper should clearly explain how LatentDiff differs from these simpler methods and why it is more effective.

3. Given the paper’s title “More Data or Better Algorithms”, a stronger experimental setup would include using a pretrained foundation model (e.g., Stable Diffusion) to synthesize new input-space data. This would test whether more data (from a large pretrained/Fondation models) can outperform better algorithms.

4. The full training pipeline is not well explained. Important details are missing:
- The architecture of the diffusion model (e.g., number of layers, hidden sizes)
- The exact conditioning mechanism for y.
- How long the diffusion model is trained, and whether it is trained jointly or separately from the encoder
- Whether only the regression head is fine-tuned after augmentation or the entire model
- How many diffusion steps T are used

5. LatentDiff aims to “generate more data,” but generative models aims  to learn/estime the data distribution (which in this case is imbalanced). Without some external source of knowledge or bias correction, it is unclear how the model can generate meaningful samples in underrepresented regions. The paper should clarify what mechanism enables the model to improve coverage of minority regions instead of reinforcing existing imbalance.

6. The experiments are not consistent or complete.
- Some baselines mentioned in the related work (e.g., ConR, VIR, SRL) are missing in the main tables.
- The set of baselines changes between datasets (e.g., ConR and BMC appear in Table 1a but not 1b).
- The California Housing results do not include baselines like LDS, FDS, or RankSim, or other DIR paper that focus on tabular data.

7. The method seems task agnostic and can also be used for imbalance classification tasks. Why the paper frame it as solving only imbalance regression tasks?

7. The paper need polishing:
- Figure 3’s legend is hard to read, and the dataset is not specified in the caption.
- The PCA plot shows clear distortion in synthetic data but this is not discussed.
- Table 2 (ablation) uses unclear terms like “No EMA” and “Noise prediction” that are never defined.
- Section 6 does not clearly refer to sub-tables (2a, 2b), and merging three tables into one caption is confusing.
- Figures use inconsistent color schemes.

**Questions:**

See Weaknesses

---

### Official Review · Reviewer_tDfL · 2025-10-31

**Soundness:** 2
**Presentation:** 2
**Contribution:** 2
**Rating:** 2
**Confidence:** 5

**Summary:**

The paper proposes LatentDiff, a latent-space conditional diffusion model for deep imbalanced regression (DIR). Instead of re-sampling, re-weighting, or feature regularization, it augments under-represented label regions by generating synthetic samples using diffusion models. The generation framework has a conditional diffusion generator, a priority-based sampling strategy, and a distributional gating mechanism to filter unrealistic samples. Experiments on IMDB-WIKI-DIR, AgeDB-DIR, STS-B-DIR, and California Housing-DIR.

**Strengths:**

The motivation is solid and exploring data augmentation for deep imbalanced regression is well-justified. The idea is technically sound, using latent diffusion augmentation to improve sample diversity in under-represented regions. Experiments on multiple DIR benchmarks demonstrate empirical effectiveness.

**Weaknesses:**

## Lack of demonstration of diffusion model
The diffusion module is the central component of this work, yet its implementation and training setup are insufficiently explained. The authors should elaborate on how the diffusion model is trained, including what dataset is used for diffusion training, and the imbalance ratio within that training data. It is unclear whether the diffusion model itself suffers from the same imbalance issue as the original dataset.
Prior studies have shown that diffusion models trained directly on imbalanced data tend to produce poor generative quality for minority regions [1, 2]. The proposed priority-based sampling may further amplify this issue by oversampling low-frequency regions. The paper need to provide visualizations or quantitative evidence of minority samples to show if they are generated with comparable fidelity.

Also I encourage the authors to qualitatively show some accepted generation and rejected generation like defined in Eq.(11).


[1] Yiming Qin et al., Class-Balancing Diffusion Models, CVPR 2023

[2] Jie Shao et al., DiffuLT: Diffusion for Long-tail Recognition Without External Knowledge, NeurIPS 2024


## Non-vision generation

The paper also reports results on STS-B, a textual similarity regression dataset. The authors should clarify what exactly is being generated here like are they generating new word embeddings, sentence pairs, or latent sentence features?

Moreover, while conditional generation is technically sound for ordinal regression tasks (e.g., age for other two datasets), the STS-B labels are continuous similarity scores that are not integers and lack natural bounds. The paper should explain how the diffusion model handles this type of conditioning.

## Wrong citations
The paper contains following incorrect citations, including wrong venues and wrong author attributions for key related works. This is not acceptable academic practice and misleads readers to locate the original references.

[1] Ranksim: Ranking similarity regularization for deep imbalanced regression.

[2] ConR: Contrastive regularizer for deep imbalanced regression.

## Experiments

The reported experimental results raise serious concerns. The baseline numbers in this paper are inconsistent with previously published results on DIR benchmarks. After reviewing several prior works, I found that reported values typically differ within a small and reasonable range. But the discrepancies here are too large to attribute to reproduction noise.
For example in the latest SOTA work (Dong et al., “Improve Representation for Imbalanced Regression through Geometric Constraints,” CVPR 2025), the many-shot MAEs in Table 1(a) are around 7.0, while this paper reports 7.5. Similar inconsistencies appear in the few-shot regions and other datasets.
The authors should explicitly explain these discrepancies (if there're differences in preprocessing?), as they undermine the credibility of the results.

**Questions:**

Please see all points in Weakness.

---

### Official Review · Reviewer_wwSx · 2025-11-07

**Soundness:** 3
**Presentation:** 3
**Contribution:** 3
**Rating:** 6
**Confidence:** 3

**Summary:**

This paper addresses Deep Imbalanced Regression (DIR), a challenging task where continuous target labels are heavily skewed, leading to poor performance on underrepresented (minority) targets. The authors argue that existing DIR solutions primarily focus on algorithmic modifications (e.g., reweighting loss functions) rather than addressing the root cause: data scarcity in minority regions.

To fill this gap, the paper proposes LatentDiff, a data-level augmentation framework that uses conditional diffusion models to generate synthetic data in the feature (latent) space of a pre-trained encoder.

Experiments on standard DIR benchmarks (IMDB-WIKI-DIR, AgeDB-DIR, STS-B-DIR) and a tabular dataset (California Housing) show that LatentDiff significantly improves performance, particularly in few-shot regions, and can be effectively combined with existing algorithmic DIR methods for state-of-the-art results.

**Strengths:**

Novel Perspective on DIR: The paper correctly identifies a gap in DIR research: the lack of effective generative data augmentation for high-dimensional data. Shifting focus from purely algorithmic corrections to tackling the fundamental data scarcity via modern generative models is a valuable contribution.

Effective Methodology: The choice to operate in feature space is pragmatically sound, offering massive computational advantages (reported as 83-174x faster generation than pixel-space diffusion) while preserving semantic manifold structure8888. The priority-based generation mechanism ($\lambda$ balancing error and scarcity) is well-motivated for the specific challenges of regression, where simple inverse-frequency sampling might not be optimal9.

**Weaknesses:**

Dependency on Backbone Quality: LatentDiff operates on features from a pre-trained encoder (e.g., ResNet-50 trained on the imbalanced data). If the initial encoder has collapsed representations for minority regions due to extreme imbalance, the diffusion model might just learn to generate realistic collapsed features. While the results suggest this isn't a fatal flaw here, it is a fundamental limitation that warrants more discussion.

Sensitivity to Dataset Scale: The authors candidly note that gains are more modest on smaller datasets (AgeDB-DIR, 15% gain) compared to larger ones (IMDB-WIKI-DIR, 46% gain). This suggests the method might struggle in highly data-scarce regimes where it is arguably most needed, as it requires enough data to learn a meaningful feature distribution to sample from.

**Questions:**

Iterative Training: Have you explored an iterative approach where the backbone encoder is fine-tuned on the augmented data, and then the diffusion model is re-trained on the improved feature space? This might alleviate the dependency on the initial biased encoder.

Backbone Pre-training: Was the ResNet-50 backbone pre-trained on ImageNet, or trained from scratch on the imbalanced datasets? If pre-trained, how much does the general feature quality of ImageNet contribute to the success of modeling minority regions compared to the diffusion augmentation itself?


Failure Modes: In the AgeDB dataset where gains were smaller, did you observe any specific characteristics of the minority samples (other than just quantity) that made them harder to model with diffusion compared to IMDB-WIKI?

---

### Note · Authors · 2025-11-16

**Comment:**

We would like to sincerely thank you for the time and care you devoted to reviewing our manuscript. After carefully reflecting on your comments and suggestions, we have decided to withdraw the paper at this stage in order to further develop and refine the work. Your feedback has been very valuable to us, and we will draw on it closely as we revise and extend this research.

**Withdrawal Confirmation:**

I have read and agree with the venue's withdrawal policy on behalf of myself and my co-authors.